# ANYTIME DENSE PREDICTION WITH CONFIDENCE ADAPTIVITY

**Zhuang Liu**[1]* **Zhiqiu Xu**[1] **Hung-Ju Wang**[1] **Trevor Darrell**[1] **Evan Shelhamer**[2]†

[1]University of California, Berkeley     [2]Adobe Research

## ABSTRACT

Anytime inference requires a model to make a progression of predictions which might be halted at any time. Prior research on anytime visual recognition has mostly focused on image classification. We propose the first unified and end-to-end approach for anytime dense prediction. A cascade of "exits" is attached to the model to make multiple predictions. We redesign the exits to account for the depth and spatial resolution of the features for each exit. To reduce total computation, and make full use of prior predictions, we develop a novel spatially adaptive approach to avoid further computation on regions where early predictions are already sufficiently confident. Our full method, named anytime dense prediction with confidence (ADP-C), achieves the same level of final accuracy, and meanwhile significantly reduces total computation. We evaluate ADP-C on Cityscapes semantic segmentation and MPII human pose estimation: our method enables anytime inference without sacrificing accuracy while also reducing the total FLOPs of its base models by 44.4% and 59.1%. We compare with anytime inference by deep equilibrium networks and feature-based stochastic sampling, showing that ADP-C dominates both across the accuracy-computation curve. Our code is available at https://github.com/liuzhuang13/anytime.

## 1 INTRODUCTION

Deep convolutional networks (Krizhevsky et al., 2017; He et al., 2016) achieve high accuracy but at significant computational cost. Their computational burden hinders deployment, especially for time-critical or low-resource use cases that for instance require interactivity or inference on a mobile device. This efficiency problem is tackled by special-purpose libraries (Chetlur et al., 2014), compression by network pruning (Han et al., 2015; Li et al., 2016; Liu et al., 2019), quantization (Rastegari et al., 2016; Jacob et al., 2018), and distillation (Hinton et al., 2015; Romero et al., 2014). These solutions accelerate network computation but the entire network must still be computed; however, a prediction may be needed sooner. Time constraints vary, but the inference time of a standard deep network does not.

*Anytime* inference (Fig. 1) mitigates this issue by bringing flexibility to model computation. An anytime algorithm (Dean & Boddy, 1988) gradually improves its results as more computation time is given. It can be interrupted at any point during its computation to return a result as system or user requirements demand. In this way, the time to the first output is reduced while the quality of the last output is preserved.

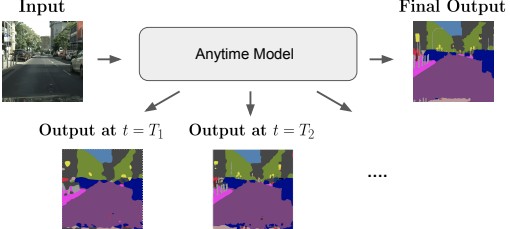

Figure 1: Anytime inference produces a progression of outputs.

An anytime model makes a progression of predictions between the first and last. This progression continues if time remains, or halts if it is either already satisfactory or out of time. For example, consider a user on a mobile device: an approximate result could be returned earlier if there is urgency, or the user could monitor the sequence of predictions as time goes by and stop the model

---

*Part of work done during an internship at Adobe Research.
†Work done at Adobe Research; the author is now at DeepMind.

once it is good enough. Note that anytime inference differs from *adaptive* or *dynamic* inference (Veit & Belongie, 2018; Wu et al., 2018; Wang et al., 2018a) where the *model* decides how much to compute instead of an *external* decision.

Prior research has explored anytime inference by feature selection (Karayev et al., 2014) or ensembling models through boosting (Grubb & Bagnell, 2012). For end-to-end neural network models, research has focused on classification for anytime inference or adaptive inference. In particular, the multi-scale dense network (Huang et al., 2017) is an architecture for resource-efficient classification. The attraction of anytime inference is not limited to classification however, and the additional computation required for dense prediction tasks makes it even more desirable. For instance, an autonomous driving system may demand swifter reaction time for safety in the presence of pedestrians, and so an anytime semantic segmentor might sooner recognize their presence. In addition to urgency, an anytime segmentor could help efficiency, by not further processing already confident predictions of street pixels and therefore save power.

In this work, we develop the first single-model anytime approach for dense prediction tasks. We adopt an early exiting framework, where multiple predictors branch off from the intermediate stages of the model. The exits are trained end-to-end (both the original exit and intermediate exits), and during inference each provides a prediction in turn. To compensate for differences in depth and spatial dimensions across stages, we redesign the predictors for earlier exits. For each exit, we choose an encoder-decoder architecture to enlarge receptive fields and smooth spatial noise.

Exits might suffice for anytime image classification, but dense prediction tasks have spatial structures. Simple regions may need less processing while complex ones need more. Standard inference applies an equal amount of computation at every pixel without taking advantage of spatial structure. Our spatially adaptive anytime inference scheme decides whether or not to continue computation at each exit and position. We mask the output of each exit by thresholding the confidence of its predictions: the remaining computation for sufficiently confident pixels is then reduced (Fig. 2). For each masked pixel, its prediction will be persisted in the following exits, as it is already sufficiently confident. In the following layers, the features for the masked pixel will be interpolated, rather than convolved, and therefore reduce computation. The confidence measure can depend on the task, e.g., in segmentation, it could be the entropy of class predictions. This *confidence adaptivity* can substantially reduce the total computation while maintaining accuracy.

We experiment with two dense prediction tasks: Cityscapes semantic segmentation and MPII human pose estimation. Redesigning the exits and including confidence adaptivity significantly improves across accuracy-efficiency operating points. Our full approach, named anytime dense prediction with confidence (ADP-C), not only makes anytime predictions, but its final predictions achieve the same level of accuracy as the base model, with 40-60% *less* total computation. For analysis, we visualize predictions and confidence adaptivity across exits, and ablate design choices for the exits and masking.

## 2 APPROACH

**Anytime Setting** In an anytime inference setting, the user can stop the inference process based on the input or a current event. Thus the computation budget for each instance $x$ could be time or input-dependent. We use $B(x, t)$ to denote the computation budget assigned for instance $x$ at time $t$, where the time variable $t$ models events that can change the budget. $B(x, t)$ could be independent of $x$, i.e., the budget only depends on the time $t$, for example if a model on a server is asked to make predictions with less budget during high-traffic hours; $B(x, t)$ can also be independent of $t$, meaning the budget is only decided by input $x$, regardless of external events. The output of the anytime model depends on the budget given, and we denote it as $f(x, B(x, t))$. Assuming $L$ is the task loss and $y$ is the ground truth, the per-instance loss is $L(f(x, B(x, t)), y)$. This leads to the expected training loss to be $\mathbb{E}_{(x,y) \sim (X,Y), t \sim T}[L(f(x, B(x, t)), y)]$, where $(X, Y)$ is the input-output joint distribution and $T$ is the distribution modeling the time or event variable.

**Early Exiting.** Next, we introduce the early exiting framework which has been used in prior works (Huang et al., 2017; Teerapittayanon et al., 2016) for anytime prediction. Standard convolutional networks only have one prediction "head" at its final stage. The network takes the input $x$, forwards it through intermediate layers, and finally outputs the prediction at its head. The concrete form of

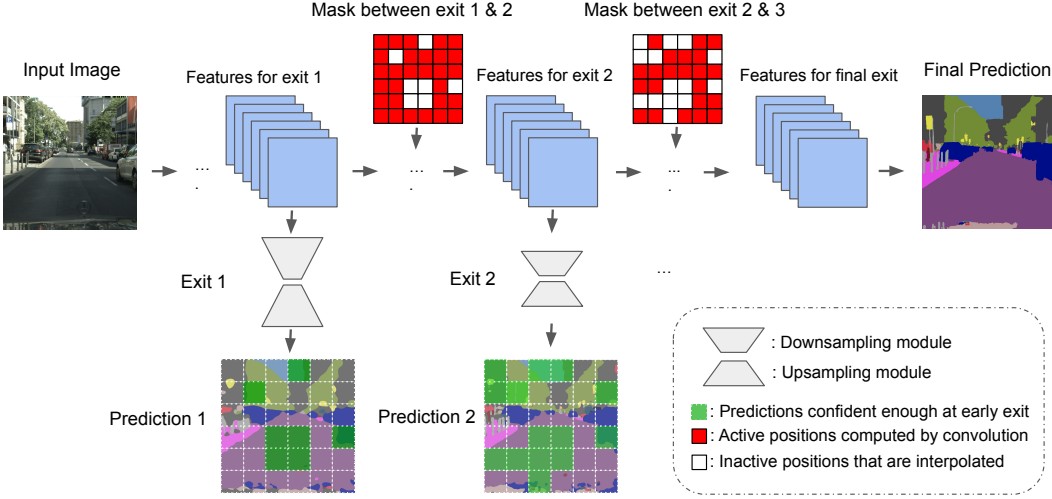

Figure 2: Our anytime dense prediction with confidence (ADP-C) approach. We equip the model with intermediate exits for anytime inference. We redesign each exit with encoder-decoder architecture to compensate for spatial resolution across model stages. At each exit's output, sufficiently confident predictions (green squares) are identified to skip further computation in the following layers.

the head depends on the task. For dense prediction, the head is usually one or multiple convolutions that output spatial maps representing pixel-wise predictions.

To obtain an anytime model, we attach multiple heads to the network, branching from its intermediate features (Fig. 2). We call these additional heads *early exits*, since they allow the network to give early predictions and stop the inference at the current layer. Suppose we add $k$ early exits at intermediate layers with layer indices $l_1 \ldots, l_k$. We denote the intermediate features at these layers $F_{l_1}(x) \ldots, F_{l_k}(x)$, and the functions represented by the early exits $E_1 \ldots, E_k$. Note that $E_i$s may be of the same form but they do not share weights. The early prediction maps can be denoted as $\hat{y}_i = E_i(F_{l_i}(x)), i = 1 \ldots k$. Together with the original final prediction $\hat{y}_{k+1}$, the total loss is:

$$L_{total} = \sum_{i=1}^{k+1} w_i L(\hat{y}_i, y) \tag{1}$$

where $w_i$ is the weight coefficient at exit $i$. The original network, together with the added exits, will be trained end-to-end to optimize this total loss function. In experiments, we set all weights equal to 1. This corresponds to the minimization of the expected loss in Sec. 2 when the exiting probabilities at all exits are equal. We find this to be a simple yet effective scheme.

For anytime inference, as the network propagates features through its layers, if the computation budget is reached or the user asks the model to stop, it will output the latest $\hat{y}_i$ that is already computed. Similar early exiting strategies have been used in resource-efficient image classification (Teerapittayanon et al., 2016; Huang et al., 2017), but dense prediction tasks require further steps detailed in the following subsections.

**Head Redesign.** Typical convolutional networks have a hierarchical structure that begins with shallow, fine, and more local features and ends with deep, coarse, and more global features. These deeper features represent more image content by their larger receptive fields. For dense prediction, upsampling is done within the network to restore lost resolution during downsampling, and ensure precise spatial correspondence between the input and the output. This upsampling can be accomplished in a few (Long et al., 2015) or many (Zhao et al., 2017) layers, but no matter the architecture, the network learns its most local features in its earliest layers. This presents a challenge for the earliest exits, since these features are limited in depth and receptive field. Making direct predictions at these exits with the typical 1×1 convolution head produces spatially noisy and inaccurate results.

To compensate for these lacking early features, we redesign the prediction heads for the exits $E_i$. Each $E_i$ first downsamples its input features $F_{l_i}(x)$, through a series of pooling and 1×1 convolution layers. Each pooling operation halves the spatial resolution, increasing its output's receptive fields. The following convolution provides the opportunity to learn new coarser-level features, specifically

for that exit's prediction. After several (denoting this number as $D$) "pool-conv" layers, we upsample the features back to the original output resolution, with an equal number ($D$) of bilinear interpolation and $1 \times 1$ convolution layers. The output of this "interpolate-conv" sequence will be the prediction $\hat{y}_i$ at this exit. This is important for ensuring spatial accuracy for pixel-level dense prediction tasks. Our redesigned exits are essentially small "encoder-decoder" modules (Fig. 2), where the encoder downsamples the features, the decoder upsamples them back.

The downsampling ratio at each exit is determined by $D$, the number of consecutive "pool-conv" layers. Intuitively, features at earlier layers are more fine-level, and the exit branching from them can potentially benefit from more downsampling. In experiments, we use an encoder with $D = N - i$ downsampling operations at exit $i$, where $N$ is the total number of exits, including the original last exit. Empirically we find this strategy works well, and alternative strategies are compared in Sec. 4.

Finally, in all early exits, the first convolution will transform the number of channels to a fixed number for all exits. By setting the channel width relatively small, we can still save computation while adding layers with this redesigned encoder-decoder head structure.

**Confidence Adaptivity.** For dense prediction tasks, any early prediction $\hat{y}_i$ is a spatial map consisting of pixel-wise predictions at each position. While most convolution networks spend an equal amount of computation at each input position, it is likely that recognition at some regions are easier than others, where the network can make predictions with high confidence even at earlier exits. For instance, the inner part of a large sky segment may be easy to recognize, whereas the boundary between the bicycle and the person riding it may need more careful delineation.

Once an early prediction is made, we can inspect the "confidence" at each position. As an example, for semantic segmentation, the maximum probability over all classes can serve as a confidence measure. If the confidence has passed a pre-defined threshold at certain positions (green squares on predictions in Fig. 2), we may decide these predictions are likely to be correct, and not continue the computation of further layers at this position. Suppose the pixels of the early prediction $\hat{y}_i$ are indexed by $p$, we form a mask $M_i$:

$$M_i(p) = \begin{cases} 0, & \text{if Confidence}(\hat{y}_i(p)) \geq \text{Threshold} \\ 1, & \text{otherwise} \end{cases} \tag{2}$$

For any convolution layer between exit $i$ ($E_i$) and the next exit $i + 1$ ($E_{i+1}$), we could choose whether to perform or skip computation at position $p$ based on the mask (Fig. 2). Assuming $C$ is a convolution layer with input $f_{in}$, then by applying the mask, the output $f_{out}$ at position $p$ becomes:

$$f_{out}(p) = \begin{cases} C(f_{in})(p), & \text{if } M_i(p) = 1, \\ 0, & \text{if } M_i(p) = 0. \end{cases} \tag{3}$$

If $C$'s output and the mask $M_i$ do not share the same spatial size, we interpolate $\hat{y}_i$ in Eqn. 2 to the size of $C$'s output, so that the mask $M_i$ is compatible with $C$ in Eqn. 3.

The output $f_{out}$ could be sparse, with many positions being 0. This could potentially harm further convolutional computation. To compensate for this, we spatially interpolate these positions from their neighbors across all channels, using a similar approach as in (Xie et al., 2020). Denoting the interpolation operation as $I$, the final output feature $f_{out}^*$ is

$$f_{out}^*(p) = \begin{cases} f_{out}(p), & \text{if } M_i(p) = 1, \\ I(f_{out})(p), & \text{if } M_i(p) = 0. \end{cases} \tag{4}$$

Here, the value of $I(f_{out})(p)$ is a weighted average of all the neighboring pixels centered at $p$ within a radius $r$:

$$I(f_{out})(p) = \frac{\sum_{s \in \Omega(p)} W_{(p,s)} f_{out}(s)}{\sum_{s \in \Omega(p)} W_{(p,s)}} \tag{5}$$

where $s$ indexes $p$'s neighboring pixels and $\Omega(p) = \{s | \|s - p\|_\infty \leq r, s \neq p\}$, the neighborhood of $p$. We set radius $r = 7$ in our experiments. $W_{(p,s)}$ is the weight assigned to point $s$ for interpolating at $p$, for which we use the RBF kernel, a distance-based exponential decaying weighting scheme:

$$W_{(p,s)} = \exp\left(-\lambda^2 \|p - s\|_2^2\right) \tag{6}$$

with $\lambda$ being a trainable parameter. This indicates that the closer $s$ is to $p$, the larger its assigned weight will be. Note that masked-out features ($M_i(p) = 0$) still participate in the interpolation process as inputs with values of 0.

Replacing filtering by interpolation at these already confident spatial locations ($M_i(p) = 0$) could potentially save a substantial amount of computation. The mask $M_i$ will be used for all convolutions between exit $i$ and $i + 1$, including the convolutions inside exit $i + 1$. Once the forward pass arrives at the next exit, to make the prediction $\hat{y}_{i+1}$, the last prediction at positions where $M_i(p) = 0$ will be carried over, having already been deemed confident enough at the last exit and having been skipped during further computations. This means:

$$\hat{y}_{i+1}(p) = \begin{cases} E_{i+1}(F_{l_{i+1}}(x)), & \text{if } M_i(p) = 1, \\ \hat{y}_i(p), & \text{if } M_i(p) = 0. \end{cases} \tag{7}$$

The network then calculates a new mask $M_{i+1}$ based on $\hat{y}_{i+1}$, and uses it to skip computation going forward. The process continues until we reach the final exit.

In summary, we incorporate spatial *confidence adaptivity* into the early exiting network, by not filtering at spatial locations that are already sufficiently confident in the latest prediction. At these positions interpolation is used instead, at much reduced computational cost, to avoid excessive sparsity. Unless otherwise specified, confidence adaptivity is used in both training and inference. We dub our full approach as anytime dense prediction with confidence (ADP-C).

## 3 EXPERIMENTS

We evaluate ADP-C with two dense prediction tasks: semantic segmentation and human pose estimation. Our experiments are implemented using PyTorch (Paszke et al., 2019).

**Architectures.** We use the High-Resolution Network (HRNet) (Wang et al., 2020) architecture as our base model. HRNet is a multi-stage architecture, where each stage adds lower-resolution/larger-scale features. Specifically, we adopt the standard HRNet-{W48,W32} models and the smaller HRNet-W18 model. HRNet-W48 is state-of-the-art for semantic segmentation and HRNet-W32 is suitable for pose estimation. HRNet-W18 is highly efficient and has been shown to outperform other efficient networks (Zhao et al., 2018; Sandler et al., 2018) in its accuracy-efficiency tradeoff. The 48/32/18 denotes the number of channels in the bottleneck of the first stage. The original head for HRNet before our redesigning is two consecutive $1 \times 1$ convolutions. We attach three exits, one at the end of each stage before the final prediction. We follow the training/evaluation protocol and hyperparameters of the reference HRNet implementation at (Sun et al., 2019; Wang et al., 2020) (except that our models include a loss at each exit). Please see Appendix C for more training details.

**Baselines. 1. HRNet**: we compare with a standard HRNet that has only one (final) exit with the same backbone architecture. The standard HRNet is not anytime, so we focus on comparing it with our anytime model's final exit. **2. MDEQ** (Bai et al., 2020) is a recent deep *implicit* model, which achieves competitive performance on vision tasks without stacking explicit layers, but rather solves an optimization problem for inference. Its representation $z^*$ is an equilibrium point of its learned transformation $f(z; x)$, i.e., $f(z^*; x) = z^*$ where $x$ is the input. The representation is obtained by iteratively solving the equation $f(z; x) = z$, for which the quality of the solution improves with more iterations. The converged representation is then decoded into a prediction. We examine anytime prediction with the MDEQ by decoding intermediate iterates of the representation. To the best of our knowledge, this is the first study of anytime implicit modeling, as (Bai et al., 2020) only reports the predictions of implicit models at equilibrium, and does not produce or inspect intermediate predictions. We use the "small" version of the MDEQ (Bai et al., 2020) and the 4th, 6th, 8th, and 10th iterations of its equilibrium optimization to bound the amount of computation and align its iterations with our architecture's stages. **3. Feature-Based Stochastic Sampling:** we follow Xie et al. (2020) in using internal features to predict masking positions, with the Gumbel-Softmax trick (Jang et al., 2016) used for sampling. We use a $3 \times 3$ convolution upon the features for exits (Fig. 2) for mask prediction until the next exit. During training, an $L_1$ sparsity regularization is applied on the stochastic continuous mask values, and only during inference the mask values are discretized. The interpolation procedure is the same as in our method. This baseline is evaluated on HRNet-W18.

|  | | Accuracy (mIoU) | | | | | Computation (GFLOPs) | | | | |
|---|---|---|---|---|---|---|---|---|---|---|---|
| Method / Output | | 1 | 2 | 3 | 4 | Avg | 1 | 2 | 3 | 4 | Avg |
| One-exit | HRNet-W48 (Wang et al., 2020) | - | - | - | 80.7 | - | - | - | - | 696.2 | - |
| Baselines | MDEQ-Small (Bai et al., 2020) | 17.3 | 38.7 | 65.5 | 72.4 | 48.5 | 521.6 | 717.9 | 914.2 | 1110.5 | 816.0 |
|  | EE (HRNet) | 34.3 | 59.0 | 76.9 | 80.4 | 62.7 | 48.4 | 113.4 | 388.9 | 722.2 | 318.2 |
| Ours | EE + RH (HRNet) | 44.6 | 60.2 | 76.6 | 79.9 | 65.3 | 41.9 | 105.6 | 368.0 | 701.3 | 304.2 |
|  | ADP-C: EE + RH + CA (HRNet) | 44.3 | 60.1 | 76.8 | **81.3** | **65.7** | 41.9 | 93.9 | 259.3 | **387.1** | **195.6** |

Table 1: Accuracy (mIoU) and inference computation (GFLOPs) for Cityscapes semantic segmentation with four exits. Our approach achieves higher accuracy in less computation than the HRNet and MDEQ baselines across exits. Early exiting (EE) makes progressive predictions. Redesigned heads (RH) improve early predictions (exits 1 and 2). Confidence Adaptivity (CA) reduces computation.

## 3.1 SEMANTIC SEGMENTATION

The Cityscapes dataset (Cordts et al., 2016) consists of 2048×1024 images of urban street scenes with segmentation annotations of 19 classes. We train the models with the training set and report results on the validation set. The accuracy metric is the standard mean intersection-over-union (mIoU %), and the computation metric is the number of floating-point operations (FLOPs). Anytime inference improves with higher accuracy, less computation, and more predictions. We evaluate HRNet-W48 and HRNet-W18 for this task.

Redesigned heads (RH) use our encoder-decoder structure for exits. Since we have 4 exits in total, we repeat the downsampling operation 3/2/1 times at exit 1/2/3 to generate larger-scale features for earlier exits, as described in Sec. 2. We set the number of channels at all exits to 128/64 for HRNet-W48/W18. For confidence adaptivity (CA), we use the maximum probability among all classes as the confidence measure, and set the confidence threshold in Eqn. 2 to be 99.8% based on cross-validation. For CA, the computation for each input can differ, so we report the average FLOPs across all validation images at each exit.

The results for HRNet-W48 are shown in Table 1. We observe that our early exiting model based on HRNet-W48 outperforms the MDEQ model by a large margin, with significantly less FLOPs at each exit. With RH, we achieve notable accuracy gain in early predictions, especially at the first exit (more than 10%), with roughly the same computation. With CA added, we arrive at our full ADP-C method (RH + CA), which maintains roughly the same accuracy as the RH model but reduces the total computation at exits 3 and 4. Interestingly ADP-C has slightly higher mIoU at the final exit (81.3 vs. 80.7) with 44.4% less total computation (387.1 vs. 696.2 GFLOPs) compared to the base HRNet. This is possibly due to a potential regularization effect of confidence adaptivity: computing fewer intermediate features exactly may prevent overfitting.

The same results are plotted in Fig. 3 (left). The plot shows accuracy ($y$-axis) and computation ($x$-axis) tradeoffs: points to the upper left indicate better anytime performance. The baseline HRNet is represented by a red cross, while anytime models are plotted as curves with a point for each prediction. We plot the results for the smaller HRNet-W18 model in Fig. 3 (middle). RH improves early prediction accuracy from the basic early exiting model, and CA substantially reduces computation at later exits. The full model reaches the same-level of accuracy as the baseline HRNet with much less total computation. We note that our model with confidence values as mask indicators also outperforms the feature-based mask sampling method in the accuracy-computation tradeoff, demonstrating that confidence is effective at filtering out redundant computation despite its simplicity.

Our experiments measure computation by FLOPs rather than time. Reporting FLOPs is common (Figurnov et al., 2017; Xie et al., 2020; Huang et al., 2017; Wang et al., 2018a; Liu et al., 2019) and meaningful because it is hardware independent. However, similarly to spatially adaptive computation methods (Figurnov et al., 2017; Xie et al., 2020), our model does not achieve wall clock speedup at this time due to the lack of software/hardware support for sparse convolution with current frameworks and GPU devices. To approximate CPU speedup, we conduct a profiling experiment on a multi-threading processor (specifically we measure computation time on a Linux machine with Intel Xeon Gold 5220R CPUs using 16 threads). We replace all convolutions with our implementations following (Xie et al., 2020). ADP-C on HRNet-W48 achieves 1.48× speedup compared to the non-anytime baseline, measured in end-to-end latency (wall-clock time). There is a gap between

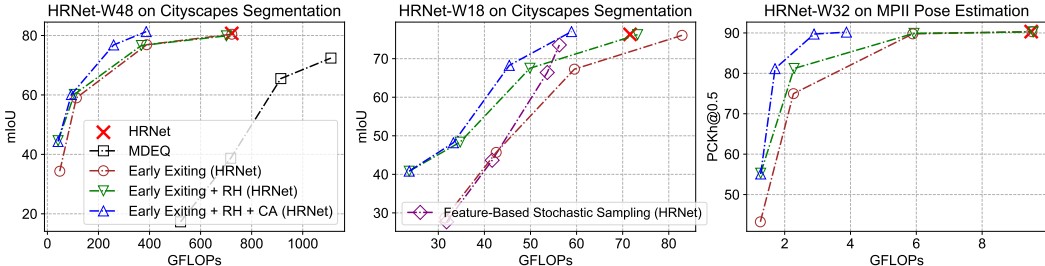

Figure 3: Accuracy ($y$) and computation ($x$) at four exits across architectures (HRNet-W48/W32/W18) and tasks (semantic segmentation and pose estimation). Anytime performance improves with higher $y$ (more accuracy) and lower $x$ (less computation). Redesigned heads (RH) boost the accuracy at early exits, while confidence adaptivity (CA) reduces computation by up to more than half. Our method outperforms baseline methods (MDEQ and Feature-based Stochastic Sampling) across the accuracy-computation tradeoff curve.

this measured time and the theoretical 1.80× speedup measured by FLOPs. ADP-C and others can benefit from ongoing and future work on efficient sparse convolutions (Graham & van der Maaten, 2017; Choy et al., 2019; Verelst & Tuytelaars, 2020; Elsen et al., 2020). We also refer readers to the Hardware Lottery (Hooker, 2021) for a discussion on how hardware compatibility affects progress in AI research. Please see the supplement for an anytime inference video where each exit is timed to the computation it requires.

## 3.2 HUMAN POSE ESTIMATION

For human pose estimation, we evaluate on the MPII Human Pose dataset (Andriluka et al., 2014) of image crops annotated with body joints collected from everyday human activities. The positions of 16 joint types are annotated for the human-centered in each crop. We report the standard metric (Andriluka et al., 2014) for MPII, the PCKh (head-normalized probability of correct keypoint) score, on its validation set. We use HRNet-W32 for this task and follow the reference settings from (Sun et al., 2019). The standard head for this task is $1 \times 1$ convolution. As in segmentation, our redesigned heads are encoder-decoder structures. The number of channels for all exits is 64.

Pose estimation task is formulated as regression. The HRNet model outputs 16 spatial feature maps, each one regressing the corresponding body joint. The only positive target for each type is coded as 1; all other points are negatives coded as 0. Unlike in segmentation, the output at each pixel is not a probability distribution, so we use the maximum value across channels as the confidence measure. A pixel is masked out if the maximum value at that position is smaller than the threshold, marking it unlikely to be a joint prediction. We choose 0.002 as the threshold by cross-validation, as a larger value makes the mask too sparse and hurts learning. The RH + CA model adopts adaptivity after 10 epochs of normal training, because nearly all outputs are too close to zero in the beginning.

| Method / Output | PCKh@0.5 | | GFLOPs | |
|---|---|---|---|---|
| | Last | Avg | Last | Avg |
| HRNet-W32 (Wang et al., 2020) | 90.33 | - | 9.49 | - |
| EE (HRNet) | 90.31 | 74.60 | 9.51 | 4.73 |
| EE + RH (HRNet) | 90.26 | 79.16 | 9.55 | 4.76 |
| ADP-C: EE + RH + CA (HRNet) | 90.20 | 79.04 | 3.88 | 2.44 |

Table 2: Accuracy (PCKh@0.5) and computation (GFLOPs) on MPII pose estimation at the last exit and averaged for all exits. Early exits (EE) make progress predictions, redesigned heads (RH) improve accuracy, and confidence adaptivity (CA) reduces computation.

Fig. 3 (right) and Table 2 show the results. We observe a similar trend to segmentation: RH improves accuracy and CA reduces FLOPs. In this case, ADP-C reduces computation by 59.1% (9.49 to 3.88 GFLOPs) while accuracy only drops by 0.13% relative to the baseline HRNet.

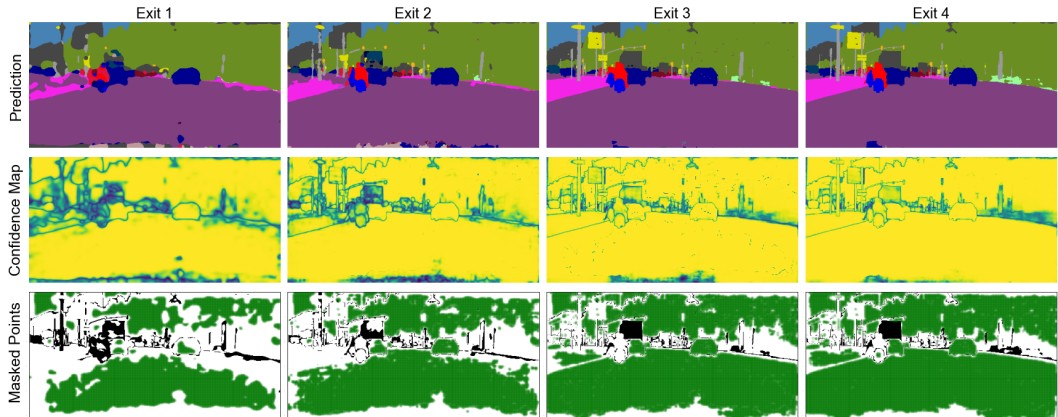

Figure 4: Top: prediction results at all exits. Middle: confidence maps, lighter color (yellow) indicates higher confidence. Bottom: correct/wrong predictions at the exit drawn as white/black. The confident points selected for masking are in green. Confidence adaptivity excludes calculation on already confident pixels (green) in early exits, mostly located at inner parts of large segments.

## 4 ANALYSIS

**Visualizations.** To inspect our anytime predictions and masking on Cityscapes, we visualize ADP-C exit results on a validation image with HRNet-W48. Fig. 4 shows the predictions, confidence maps, and computation masks across exits. With each exit, the prediction accuracy improves, especially in more detailed areas with more segments. The confidence maps are shown with high lighter/yellow and low darker/green. Most unconfident points lie around segment boundaries, and the interior of large stuff segments (road, vegetation) are already confident at early exits. This motivates the use of confidence adaptivity to avoid unnecessary further computations on these areas. For computation masks, the correct/incorrect predictions at each exit are marked white/black. Pixels surpassing the confidence threshold (99.8%) are masked and marked green. Many pixels can be masked out in this way, and each exit masks more. Most of the masked pixels are found in the inner parts of large segments or already correct areas. In fact, the masked pixels are 100% correct at all exits for this instance, which partly justifies their exclusion from later computation. The predictions at these positions are already confident and correct at early exits, and so the only potential harm of skipping their computation later is their possible effect at less confident positions. See Appendix E for more visualizations. On average, 19.3%, 38.4%, 63.0% pixels are masked out at exit 1, 2, 3, respectively.

**Downsampling at Early Exits.** In Sec. 2, we described how many consecutive downsampling operations we use at each exit, by $D = N - i$, which means we use $D = 3/2/1$ consecutive "pool-conv" layers for downsampling at exit $1/2/3$. Here we compare this strategy with $D = 1/1/1$ and $3/3/3$, where the same level of downsampling and hence the same head structure is used at all exits, on HRNet-W18. Fig. 5 (left) shows that our adopted $D = 3/2/1$ strategy obtains the highest accuracy at all exits among these choices.

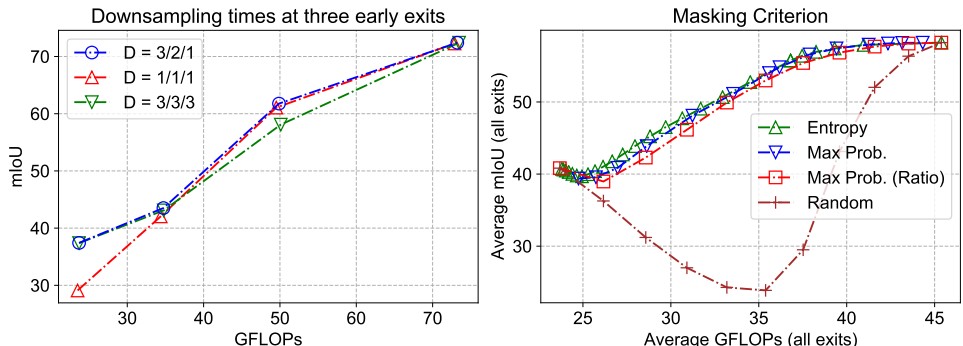

Figure 5: *Left:* comparing downsampling strategies. $D = 3/2/1$ means downsampling the features $3/2/1$ times at exit $1/2/3$. *Right:* comparison between different masking criteria.

**Masking Criterion.** We used the max probability as the confidence measure and a fixed threshold for masking. Here we consider a few alternatives. One is to mask out the top $k\%$ (by max prob.) of the pixels at each exit, regardless of their values. We also consider thresholding on the entropy of the probability distribution. In addition, we compare them with random masking. We use HRNet-W18, change the ratio or threshold for a wide range, and present the average mIoU vs. GFLOPs on all exits at Fig. 5 (right). For this ablation, adaptivity is only applied during inference.

First, we notice all three confidence criteria largely outperform random masking. For max probability, using a threshold performs slightly better than a fixed ratio, possibly because this gives the flexibility for different exits to mask out different amounts of points. Finally, we observe using entropy as the confidence measure performs similarly as using max probability, but we stick to max probability in our methods because it is trivial to compute.

## 5  RELATED WORK

**Anytime Inference.**  *Anytime* algorithms (Zilberstein, 1996; Dean & Boddy, 1988) can be interrupted at any point during computation to return a result, whose quality improves gradually with more computation time. In machine learning, anytime inference has been achieved by boosting (Grubb & Bagnell, 2012), reinforcement learning (Karayev et al., 2014), and random forests (Fröhlich et al., 2012). Anytime deep networks have been brought to bear on image classification, but not dense prediction. Branching architectures have been a common strategy (Amthor et al., 2016; Teerapittayanon et al., 2016) along with other techniques such as adaptive loss balancing (Hu et al., 2019). While there is work on the tasks of person re-identification (Wang et al., 2019) and stereo depth (Wang et al., 2018b), these techniques are task-specific, while our method applies to multiple dense prediction tasks, as we show with semantic segmentation and pose estimation. Liu et al. (Liu & He, 2016) learn a hierarchy of models for anytime segmentation, but its multiple models complicate training and testing, and require more memory. Our work instead augments the base model architecture for simplicity and efficiency. The PointRend method (Kirillov et al., 2020) outputs a initial dense prediction first and then refine it adaptively, but the predictions are all made at full depth. Its majority of the computation is spent before the first output and thus cannot be practically "anytime". Our method is the first to selectively update anytime predictions across space and layers.

**Adaptive Computation.** An *adaptive* model adjusts its computation to each specific instance during inference. For deep networks, this is often done by adjusting which layers to execute, that is, choosing which layers to run or skip. This can be done by a supervised controller (Veit & Belongie, 2018; Liu & Deng, 2017), a routing policy optimized by reinforcement learning (Wang et al., 2018a; Wu et al., 2018; Lin et al., 2017a), or other training strategies (McGill & Perona, 2017). Rather than choosing layers, *spatial adaptivity* chooses where to adjust the amount of computation across different spatial positions in the input. For example, the model could infer spatial masks for feature maps and skip computation on masked areas (Shomron et al., 2019; Dong et al., 2017; Lin et al., 2017b; Ren et al., 2018; Cao et al., 2019). Figurnov et al. (2017) maintains a halting score at each pixel and once it reaches a threshold the model will stop inference at those positions for spatially coarse tasks like classification or bounding box detection. Xie et al. (2020) stochastically sample positions for computation from an end-to-end learned sampling distribution. Li et al. (2017) convert a deep network into a difficulty-aware cascade, where earlier steps handle easier regions and later steps tackle harder regions. These spatially adaptive models reduce computation, but are not anytime: they do not make a series of predictions and cannot be interrupted.

## 6  CONCLUSION

We propose ADP-C, the first single-model anytime approach for dense visual prediction. Based on an early-exiting framework, our redesigned exiting heads and confidence adaptivity both improve the accuracy-computation tradeoff. On Cityscapes semantic segmentation and MPII pose estimation, ADP-C achieves 40%-60% FLOPs reduction with the same-level final accuracy, compared against the baseline HRNet. We further analyze confidence adaptivity with visualizations and ablate key design choices to justify our approach to anytime inference with confidence.

**Acknowledgement.** This work was supported in part by DoD including DARPA's XAI, LwLL, and/or SemaFor programs, as well as BAIR's industrial alliance programs.

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

# APPENDIX

## A  ABLATION ON INTERPOLATION RADIUS

In the main paper, we use a default radius of $r = 7$ when interpolating masked-out features (Eqn. 6 and following text). Here we demonstrate that the radius of 7 is reasonable through an ablation experiment, whose results are listed in Table 3. The experiment is conducted with HRNet-W18 on Cityscapes semantic segmentation, with both redesigned head (RH) and confidence adaptivity (CA). We observe that a radius of 7 outperforms lower ones (3 and 5) in the average mIoU slightly, with minimal FLOPs addition, due to the fact that interpolation is done channel-wise. A further increase of radius to 9 does not bring significant gain in final or average mIoU.

| Radius | Accuracy (mIoU) | | | | | Computation (GFLOPs) | | | | |
|---|---|---|---|---|---|---|---|---|---|---|
| | 1 | 2 | 3 | 4 | Avg | 1 | 2 | 3 | 4 | Avg |
| 3 | 41.06 | 48.25 | 67.56 | 75.82 | 58.17 | 23.7 | 33.0 | 44.4 | 57.0 | 39.5 |
| 5 | 40.86 | 48.01 | 67.64 | 76.21 | 58.18 | 23.7 | 33.1 | 44.6 | 57.4 | 39.7 |
| 7 (default) | 41.05 | 48.35 | 67.73 | 76.10 | 58.31 | 23.7 | 33.1 | 44.9 | 58.1 | 40.0 |
| 9 | 41.01 | 48.41 | 67.88 | 76.08 | 58.35 | 23.7 | 33.2 | 45.4 | 59.1 | 40.4 |

Table 3: Accuracy (mIoU) and inference computation (GFLOPs) for Cityscapes semantic segmentation with four exits under different settings of interpolation radius. A minor increase in mIoU and GFLOPs is observed with a larger radius.

## B  INFERENCE-ONLY CONFIDENCE ADAPTIVITY

| Adaptivity | Accuracy (mIoU) | | | | | Computation (GFLOPs) | | | | |
|---|---|---|---|---|---|---|---|---|---|---|
| | 1 | 2 | 3 | 4 | Avg | 1 | 2 | 3 | 4 | Avg |
| No Adaptivity | 44.61 | 60.19 | 76.64 | 79.89 | 65.33 | 41.9 | 105.6 | 368.0 | 701.4 | 304.2 |
| Training and Inference | 44.34 | 60.13 | 76.82 | 81.31 | 65.65 | 41.9 | 93.9 | 259.4 | 387.1 | 195.6 |
| Inference-only | 44.61 | 59.97 | 76.37 | 79.69 | 65.16 | 41.9 | 94.1 | 291.8 | 484.8 | 228.1 |

Table 4: Accuracy (mIoU) and inference computation (GFLOPs) for Cityscapes semantic segmentation with four exits under different adaptivity settings.

In the experiment section of the main paper, confidence adaptivity is used in both training and inference. Here we compare this with the setting where adaptivity is only used for inference in Table 4. We use HRNet-W48 with redesigned heads (RH) on Cityscapes for this experiment. "No Adaptivity" corresponds to the "EE + RH" row in Table 1. We observe that using adaptivity only at inference hurts the accuracy, compared with using it at both training and inference. It also increases the average FLOPs at exit 3 and 4.

## C  EXPERIMENTAL SETTINGS

For Cityscapes semantic segmentation, we follow the training settings at the official codebase (HR-Net Semantic Segmentation) of HRNet for semantic segmentation. The HRNet-W18/48 models are pre-trained on ImageNet. During training, multi-scale and flipping data augmentation is used, and the input cropping size is $512 \times 1024$. The model is trained for 484 epochs, with an initial learning rate of 0.01 and a polynomial schedule of power 0.9, a weight decay of 0.0005, a batch size of 12, optimized by SGD with 0.9 momentum. In evaluation, we use single-scale testing without flipping, with input resolution $1024 \times 2048$. For the Feature-Based Stochastic Sampling baseline, we modify the exit weights from $(1, 1, 1, 1)$ to $(0.5, 0.5, 0.5, 1)$ as we find it produces more stable masking values during training. The $L_1$ sparsity regularization on masks is set to 0.1. The mask outputs at exit $(1, 2, 3)$ have additional weight factors of $(1/3, 2/3, 1)$ to encourage more sparse features towards the end.

For MPII human pose estimation, we follow the training settings at the official codebase (HRNet Human Pose Estimation) of HRNet for pose estimation. The HRNet-32 model we use is also pre-trained on ImageNet. The image size for both training and evaluation is $256 \times 256$. The model is trained for 210 epochs, with an initial learning rate of 0.001, and decaying of 0.1 at epoch 170 and 200. The optimization is done by Adam with $\gamma_1 = 0.99, \gamma_2 = 0$, a weight decay of 0.0001, and a momentum of 0.9. The batch size is 128. In evaluation, flipping test is used.

## D PASCAL-CONTEXT RESULTS

We present results with the PASCAL-Context semantic segmentation dataset (Mottaghi et al., 2014) in Table 5. It consists of 59 segmentation classes. For the Early Exiting baseline, we use weights of (0.33, 0.33, 0.33, 1) for 4 exits, respectively, as we found increasing it to all 1 would hurt the final performance too much. The confidence threshold is set to 99.5%. We observe RH improve the accuracy in most exits. ADP-C outperforms the Early Exiting baseline in average mIoU, despite its final mIoU is worse. It also saves more than 20% FLOPs compared with the vanilla Early Exiting baseline.

| | Method / Output | Accuracy (mIoU) | | | | | Computation (GFLOPs) | | | | |
|---|---|---|---|---|---|---|---|---|---|---|---|
| | | 1 | 2 | 3 | 4 | Avg | 1 | 2 | 3 | 4 | Avg |
| One-exit | HRNet-W48 (Wang et al., 2020) | - | - | - | 51.35 | - | - | - | - | 76.5 | - |
| Baseline | Early Exiting (HRNet) | 8.78 | 18.74 | 39.82 | 50.10 | 29.36 | 5.4 | 12.6 | 43.1 | 80.1 | 35.3 |
| Ours | EE + RH (HRNet) | 14.55 | 19.42 | 39.30 | 50.32 | 30.90 | 4.5 | 11.4 | 40.2 | 77.2 | 33.3 |
| | ADP-C: EE + RH + CA (HRNet) | 14.51 | 19.18 | 39.00 | 49.27 | 30.49 | 4.5 | 11.1 | 38.1 | **62.7** | 29.1 |

Table 5: Accuracy (mIoU) and inference computation (GFLOPs) for PASCAL-Context semantic segmentation.

## E MORE VISUALIZATIONS

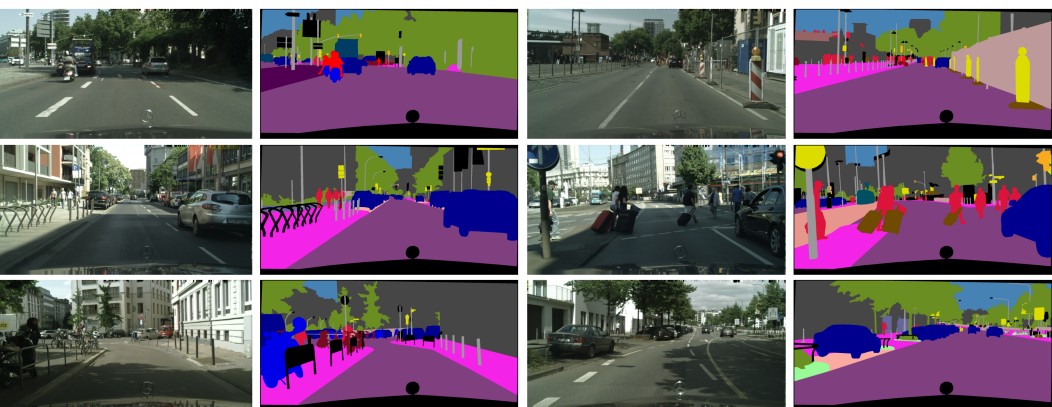

Figure 6: Input and ground truth of the example validation images visualized in Fig. 7 and Fig. 8

We present more visualizations of the same type as Fig. 4 in the main paper, in Fig. 7 and 8. Their input and ground truth are shown in Fig. 6 in the same order. We can see the same trend as discussed in the main paper still holds: the model will mask out confident points that are inside large segments (e.g., road, vegetable), which are mostly already predicted correctly in early exits.

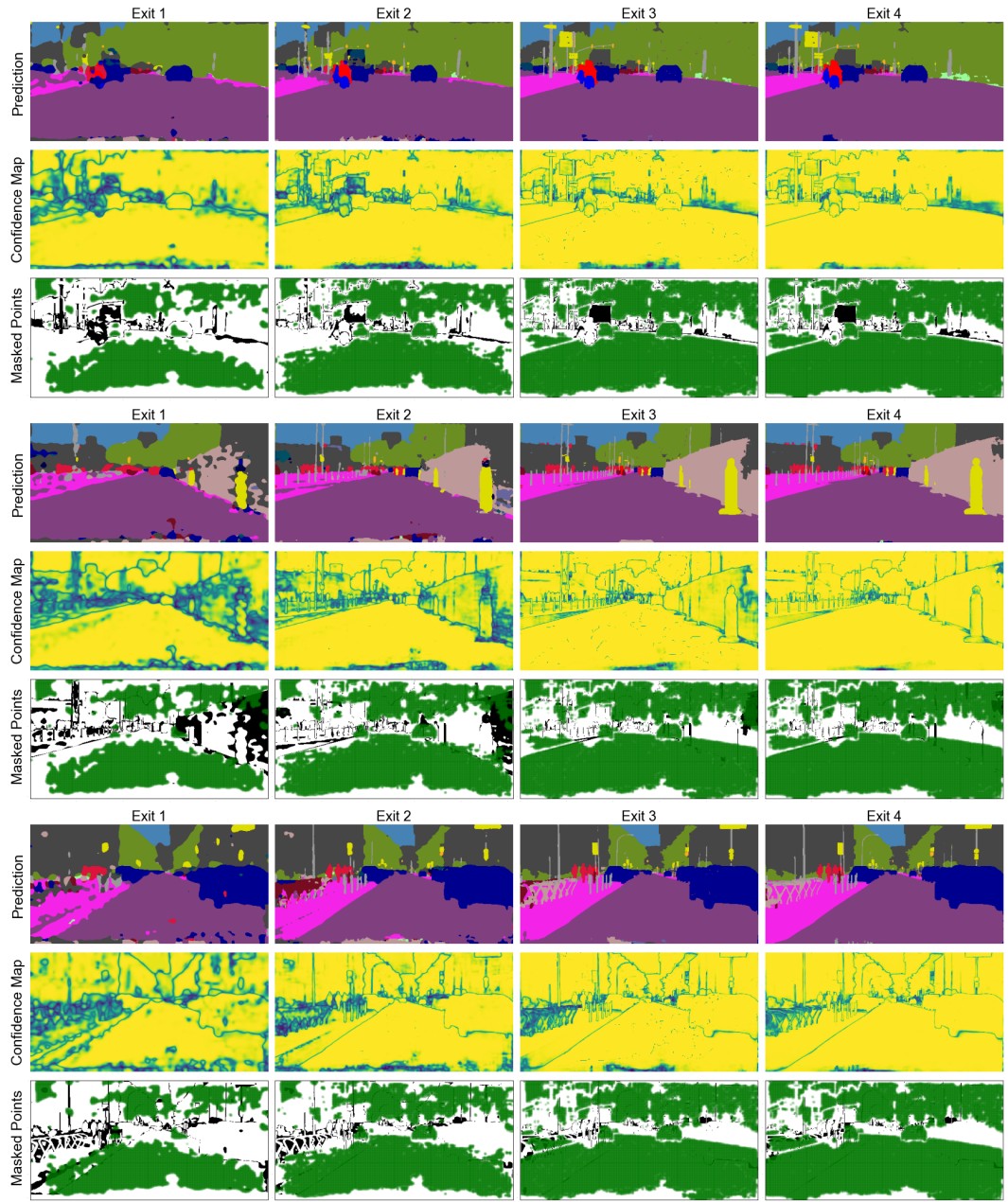

Figure 7: Top: prediction results at all exits. Middle: confidence maps, lighter color (yellow) indicates higher confidence. Bottom: correct/wrong predictions at the exit drawn as white/black. The confident points selected for masking are in green. Confidence adaptivity excludes calculation on already confident pixels (green) in early exits, mostly located at inner parts of large segments.

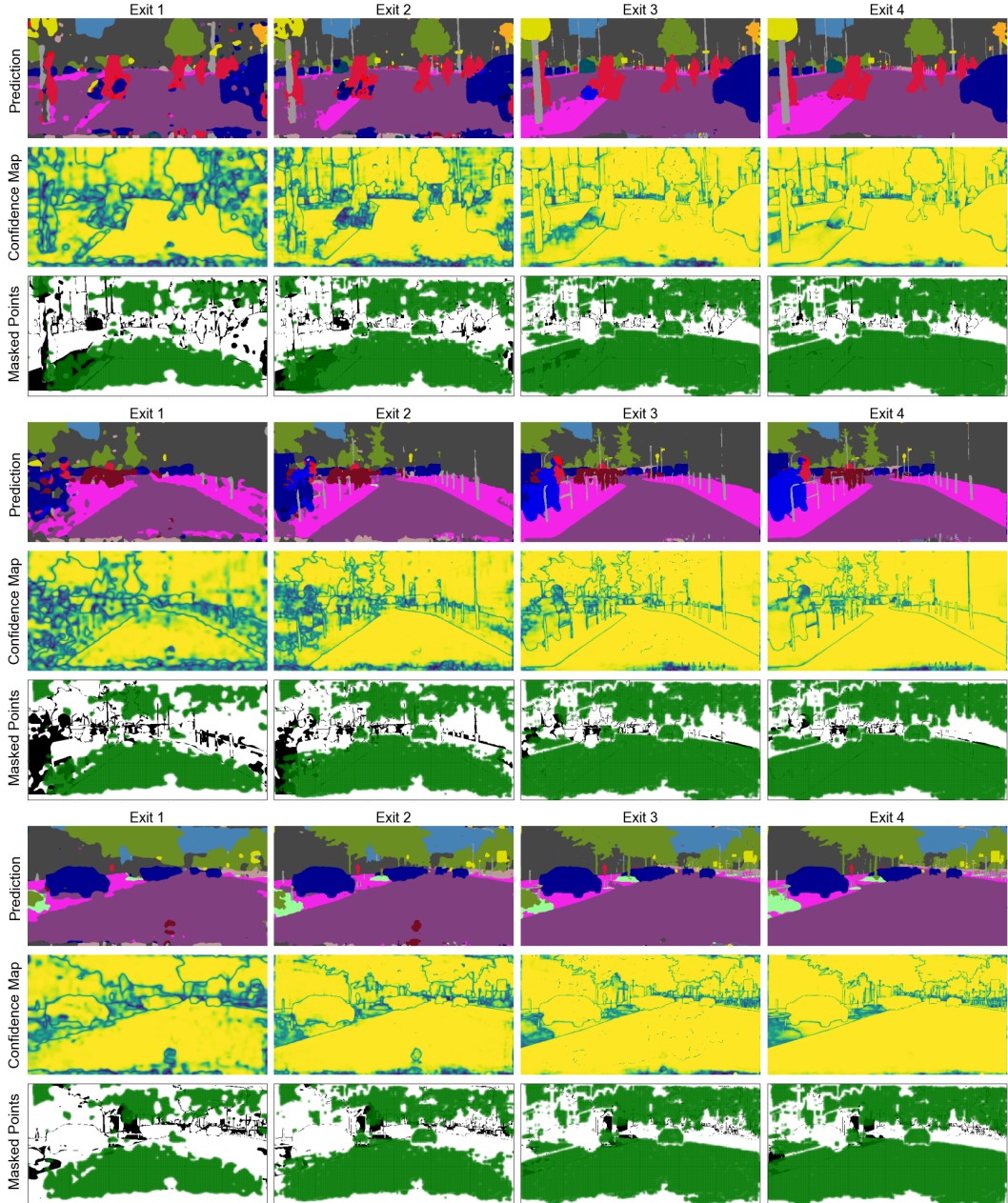

Figure 8: Top: prediction results at all exits. Middle: confidence maps, lighter color (yellow) indicates higher confidence. Bottom: correct/wrong predictions at the exit drawn as white/black. The confident points selected for masking are in green. Confidence adaptivity excludes calculation on already confident pixels (green) in early exits, mostly located at inner parts of large segments.

# F MORE MDEQ RESULTS

In the main paper, we used MDEQ's "small" model's 4th, 6th, 8th, and 10th iterations' results to align with our model's 4 exits. In this section, we provide full results on Cityscapes semantic segmentation at all iterations for both the "small" and the "XL" (extra-large) models at Table 6. The original configuration (MDEQ) sets the number of iterations to 26 and 27 for MDEQ-small and MDEQ-XL. In Fig. 9 and 10, we also provide qualitative visualization of results for the same 6 validation images from Fig. 6, with MDEQ-small at 4th, 6th, 8th, 10th, 14th, 18th, 22th, 26th iterations. With the progression of iterations, the predictions get more and more accurate.

| | MDEQ-Small | | MDEQ-XL | |
|---|---|---|---|---|
| Iteration | mIoU | GFLOPs | mIoU | GFLOPs |
| 1 | 1.1 | 227.2 | 1.9 | 1983.0 |
| 2 | 1.1 | 325.3 | 1.9 | 2861.3 |
| 3 | 9.0 | 423.5 | 8.0 | 3739.6 |
| 4 | 17.3 | 521.6 | 11.6 | 4617.9 |
| 5 | 38.7 | 619.7 | 34.1 | 5496.2 |
| 6 | 38.7 | 717.9 | 49.4 | 6374.5 |
| 7 | 61.0 | 816.0 | 58.6 | 7252.9 |
| 8 | 65.5 | 914.2 | 67.1 | 8131.2 |
| 9 | 70.1 | 1012.3 | 71.6 | 9009.5 |
| 10 | 72.4 | 1110.5 | 74.5 | 9887.8 |
| 11 | 73.8 | 1208.5 | 76.1 | 10766.1 |
| 12 | 74.6 | 1306.7 | 77.3 | 11644.4 |
| 13 | 75.1 | 1404.8 | 77.9 | 12522.7 |
| 14 | 75.5 | 1503.0 | 78.7 | 13401.0 |
| 15 | 75.8 | 1601.1 | 79.1 | 14279.3 |
| 16 | 75.9 | 1699.3 | 79.3 | 15157.6 |
| 17 | 76.1 | 1797.4 | 79.5 | 16036.0 |
| 18 | 76.2 | 1895.5 | 79.6 | 16914.3 |
| 19 | 76.2 | 1993.7 | 79.6 | 17792.6 |
| 20 | 76.2 | 2091.8 | 79.7 | 18670.9 |
| 21 | 76.2 | 2190.0 | 79.8 | 19549.2 |
| 22 | 76.1 | 2288.1 | 79.9 | 20427.5 |
| 23 | 76.2 | 2386.2 | 79.9 | 21305.8 |
| 24 | 76.3 | 2484.4 | 79.9 | 22184.2 |
| 25 | 76.4 | 2582.5 | 79.9 | 23062.5 |
| 26 | 76.5 | 2680.7 | 79.8 | 23940.8 |
| 27 | 76.5 | 2778.8 | 79.8 | 24819.1 |

Table 6: Accuracy (mIoU) and computation (GFLOPs) on Cityscapes semantic segmentation for MDEQ models, at different iterations.

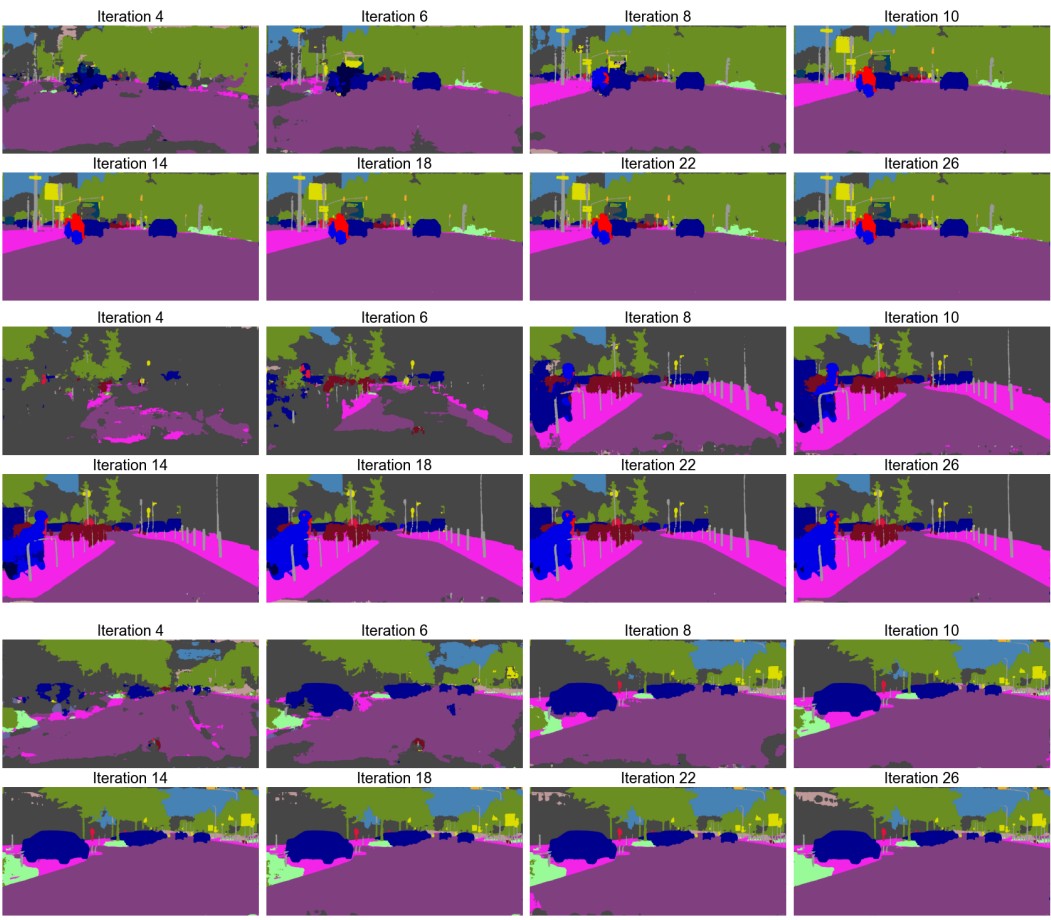

Figure 9: Cityscapes prediction results for MDEQ-Small all various iterations. Input and ground truth are in Fig. 6.

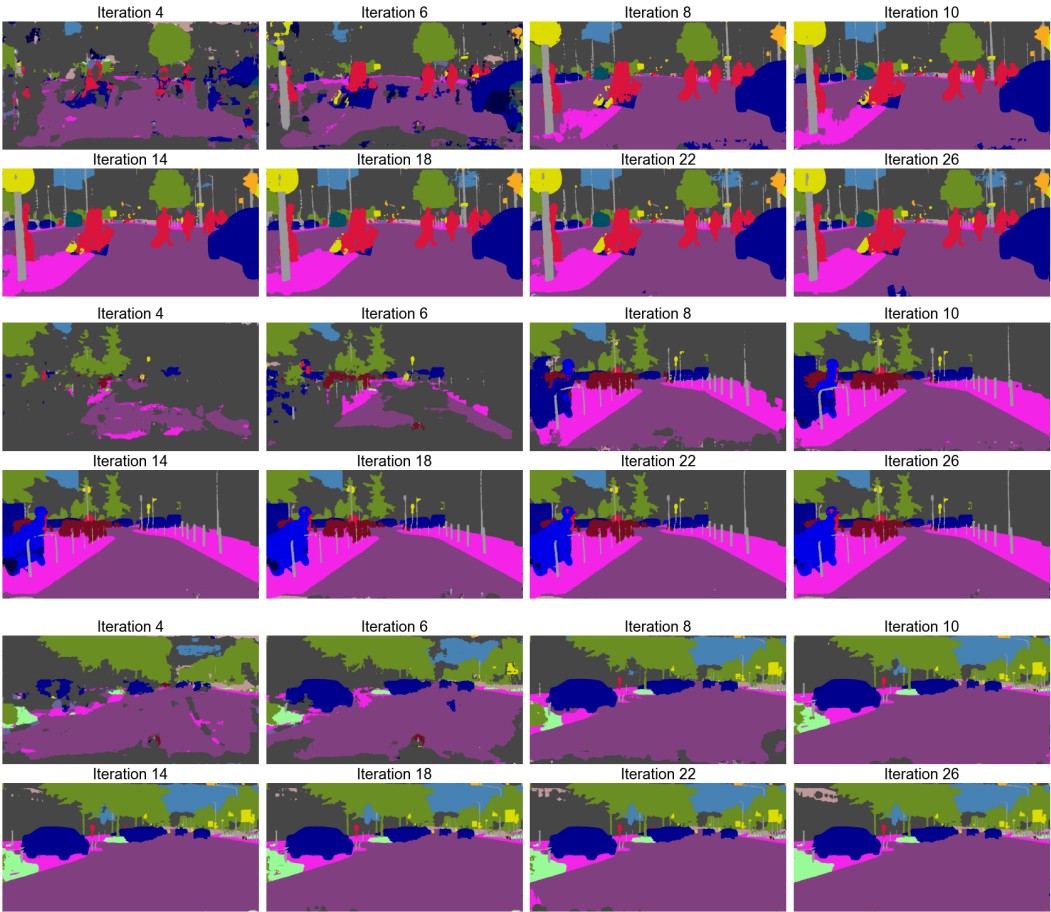

Figure 10: Cityscapes prediction results for MDEQ-Small all various iterations. Input and ground truth are in Fig. 6.

