# OpenReview forum: "Anytime Dense Prediction with Confidence Adaptivity"
_ICLR.cc/2022/Conference — ICLR 2022 Poster_

### Official Review · Reviewer_2wqr · 2021-10-27

**Correctness:** 3
**Technical Novelty And Significance:** 2
**Empirical Novelty And Significance:** 2
**Recommendation:** 6
**Confidence:** 3

**Main Review:**

Strengths:
+ The paper is well written.
+ The motivation behind the proposed problem is clear and convincing.
+ The results are reasonable, i.e., the authors are able to match or improve the HRNet baseline while reducing the computational complexity.

Weaknesses:
- I'm not convinced of the novelty of the "anytime" prediction problem. The authors might be the first to formulate it as an actual research problem. However, the main requirement of this problem is to predict segmentation maps sequentially / progressively. It seems that there have already been several much older methods that do this, i.e., "Holistically-Nested Edge Detection" (ICCV 2015), which uses deep supervision to predict dense edge maps at every stage of the network, "DeepLab: Semantic Image Segmentation with Deep Convolutional Nets, Atrous Convolution, and Fully Connected CRFs" (ICLR 2015), which also produces segmentation maps at every resolution level, etc. I'm sure there are many other methods that do this, but these are the first ones / most popular ones that come to mind. The authors in these papers don't explicitly discuss the "anytime" prediction problem, but technically one could adapt these methods to this problem quite easily.
- I'm also not convinced of the novelty of the proposed approach. The exit branches seem very similar to standard prediction branches attached to different stages of the network (as in the two papers listed above). Furthermore, the concept of confidence adaptivity shares lots of similarity with prior methods such as "PointRend: Image Segmentation as Rendering", and "Not All Pixels Are Equal: Difficulty-Aware Semantic Segmentation via Deep Layer Cascade", both of which also focus the computation of their model towards pixels that are not predicted confidently. Of course, there are subtle differences between the proposed approach and these prior methods. However, the main concepts are quite similar, which diminishes the technical novelty of the proposed approach.
- The experiments are lacking. Considering that the authors claim that they are the first ones to perform evaluations on this problem, I would expect more extensive experiments with different models/backbones (not just with HRNet, which is a bit outdated at this point). I'm not familiar with the latest state-of-the-art in semantic segmentation, but it would be useful to include at least several additional models to demonstrate the generality of the proposed approach.
- I'm confused why the accuracy in Table 1 improves when adding the  Confidence Adaptivity component. My intuition was that this should reduce the computational cost of the method, but that it should also decrease the accuracy (as we are only focusing on a subset of pixels in the future maps). Therefore, I'm confused why the accuracy would actually increase in this case.
- From Figure 5, it doesn't seem that the downsampling strategy at each exit matters that much, i.e., for three out four exits, the 3/3/3 strategy works as well as the proposed 3/2/1 scheme.


**Summary Of The Paper:**

The paper proposes a new task called "anytime" prediction, which requires a model to make a progression of predictions, which might be halted at any time. The authors then also introduce an end-to-end model for this problem. The main two components behind the proposed model are: (1) a cascade of “exits” enabling the model to make progressive predictions while taking into account accuracy vs computational cost tradeoff; (2) Confidence Adaptivity, which allows the model to focus on the less confident pixel prediction. The authors implement their approach using HRNet baseline and demonstrate improved performance and efficiency on semantic segmentation and pose prediction tasks.

**Summary Of The Review:**

I'm not convinced that the proposed problem is novel. Furthermore, considering the conceptual similarity to several prior methods (see the discussion above), I believe that the technical novelty of the paper is also quite limited. Lastly, in my opinion, the current experiments are lacking. Therefore, I would recommend rejecting the paper.

---

> ### Author Response · Authors · 2021-11-23
> **Response to Reviewer 2wqr [1/2]**
>
>
> We sincerely thank you for your constructive comments. We are encouraged that you find the motivation convincing and the paper well written. We would like to address the comments and questions below.
>
> 1. > I'm not convinced of the novelty of the "anytime" prediction problem. The authors might be the first to formulate it as an actual research problem....
>
> **We would like to clarify that, we do not claim to be the first to propose the "anytime" prediction problem setting. In related works (Section 2), we introduced multiple prior works that considered the anytime setting for various tasks.** We view our approach as the first general method for anytime **pixel-level recognition**.
>
> > "Holistically-Nested Edge Detection" (ICCV 2015), which uses deep supervision to predict dense edge maps at every stage of the network.
>
> The HED method learns different edges at different layers, and requires fusing multiple side outputs to give the final edge detection result. Each stage's output contributes to the aggregated prediction, but **each stage's output itself is not considered a final prediction**.
>
> > "DeepLab: Semantic Image Segmentation with Deep Convolutional Nets, Atrous Convolution, and Fully Connected CRFs" (ICLR 2015), which also produces segmentation maps at every resolution level
>
> DeepLab has a fusion operation (Atrous Spatial Pyramid Pooling) that **aggregates features at different scales but it only leads to one prediction. In addition, these different features are parallel, not sequential (Figure 7 in [1]), thus cannot be easily adapted to "anytime".** Many models use skip connections or multi-layer fusion operations, but these models are not necessarily easily convertible to "anytime" models.
>
> [1] DeepLab: Semantic Image Segmentation with Deep Convolutional Nets, Atrous Convolution, and Fully Connected CRFs. Chen et al. 2016. URL: https://arxiv.org/abs/1606.00915
>
>
> 2. > I'm also not convinced of the novelty of the proposed approach.... the concept of confidence adaptivity shares lots of similarity with prior methods such as "PointRend: Image Segmentation as Rendering"
>
> PointRend is a strong segmentation method. However, the majority of computation (>90%) for PointRend is spent on the initial dense prediction. The refinement that follows can be considered "progressive", but **they are all at the same full depth**. Unlike our method which outputs result in relatively evenly distributed time, one would have to wait >90% of the total time for the first prediction for PointRend, thus **PointRend is not practically "anytime"**. **We have added a short discussion on PointRend in the revision's related work section.**
>
> > "Not All Pixels Are Equal: Difficulty-Aware Semantic Segmentation via Deep Layer Cascade"
>
> The Deep Layer Cascade (DLC) in this paper is indeed highly related to our method, and **we discussed it in our related work's "Adaptive Computation" paragraph**. DLC did not report mIoU other than at the final head, and is not designed for the anytime setting. Its goal is to improve final accuracy and overall speed by adjusting the computation needed for each image.
>
>
>
> > the main concepts are quite similar, which diminishes the technical novelty of the proposed approach.
>
> **We acknowledge that the idea of spending more computation on difficult regions and less on easy regions is quite prevalent, as discussed extensively in the "Adaptive Computation" paragraph of the related work section. Nevertheless, its application in an anytime setting was not previously explored.**
>
> In addition, **we would like to clarify that we do not view EE as our new contribution or technical novelty**. Our contributions include the **confidence adaptivity mechanism**, which outperformed the feature-based stochastic sampling method in Xie et al.; the **redesigned exits**, which improved early exit results significantly; and **experimental results on recent alternatives** such as deep equilibrium networks (MDEQ), and feature-based stochastic sampling. This was indeed not communicated clearly in our original submission. **We have updated our paper, moving the EE result row in Table 1 from "Ours" to "Baselines", merging "Anytime Setting" and "Early Exiting" paragraphs in Section 3, and mentioning again EE was used in prior works before we introduce it.** We hope this can help make our contributions more clear.

---

> > ### Author Response · Authors · 2021-11-23
> > **Response to Reviewer 2wqr [2/2]**
> >
> >
> > 3. > Considering that the authors claim that they are the first ones to perform evaluations on this problem ...
> >
> > As noted by our response to 1., we only view our approach as the first general anytime method for pixel-level recognition.
> >
> >
> > > The experiments are lacking... I would expect more extensive experiments with different models/backbones (not just with HRNet, which is a bit outdated at this point).
> >
> >
> > **As noted by Reviewer GYrC, HRNet is still considered current state-of-the-art for semantic segmentation.** For example, please see the Cityscapes leaderboard at https://paperswithcode.com/sota/semantic-segmentation-on-cityscapes, where several top entries are still based on HRNet. **We chose HRNet as our backbone because it is (relatively) simple, effective, and quite widely used.** The TPAMI version of HRNet [2] was published in 2020, thus we don't view it is necessarily outdated.
> >
> > We are experimenting with PSPNet [1] to test our method on another architecture. We will add the result to our next version, and post an update response if we manage to get the results before the discussion period ends (Nov 29). PSPNets are quite different from HRNets so it will take some time and effort. We also welcome suggestions on which ConvNet-based architecture(s) to further test our method on.
> >
> > Additionally, **we added results on the PASCAL Context dataset [3] in Appendix C.**
> >
> > [1] Pyramid scene parsing network. Zhao et al. CVPR 2017. \
> > [2] Deep High-Resolution Representation Learning for Visual Recognition. Wang et al. TPAMI 2020. URL: https://ieeexplore.ieee.org/document/9052469 \
> > [3] The Role of Context for Object Detection and Semantic Segmentation in the Wild. Mottaghi et al. CVPR 2014.
> >
> >
> >
> > 4. > I'm confused why the accuracy in Table 1 improves when adding the Confidence Adaptivity component. My intuition was that this should reduce the computational cost of the method, but that it should also decrease the accuracy (as we are only focusing on a subset of pixels in the future maps).
> >
> > In Table 1, we did empirically observe the improvement in accuracy while reducing computation cost. **As we hypothesized in the paper, this might be due to the regularization effect from masking**, as the redundancy and fitting power of the model is reduced by the confidence-adaptive masking operation. We also note **this accuracy increase did not happen for every task/model we evaluated**, e.g., in pose estimation.
> >
> > Additionally, what portion of pixels each exit masks out can be adjusted by the global hyperparameter of confidence threshold. In our Cityscapes experiment, it is set to 99.8%. If we set this threshold to be low enough, the accuracy will start to be hurt. In our paper, we used a value that gives a decent final performance.
> >
> > 5. > From Figure 5, it doesn't seem that the downsampling strategy at each exit matters that much, i.e., for three out four exits, the 3/3/3 strategy works as well as the proposed 3/2/1 scheme.
> >
> > From Figure 5, we can see **the 3/2/1 scheme's mIoU outperforms the 3/3/3 scheme at the third exit, by a quite wide margin. 3/2/1 also uses less parameters/FLOPs than the 3/3/3 scheme.** The head design of the 3/2/1 scheme **only differs from 3/3/3 at two exits (2nd and 3rd), and at one out of these two exits (50%), we achieve a big improvement,** which we think justifies the head redesign. We aim to optimize the accuracy for all exits, thus 3/2/1 is still our preferred choice. **We also would like to note the original head design is 1/1/1 instead of 3/3/3, and our 3/2/1 outperforms 1/1/1 at every exit.**
> >
> > We'd like to thank you again for the valuable feedback and pointed references. We hope our response and revision could address your concerns.

---

> > > ### Author Response · Authors · 2021-11-29
> > > **More architecture result update**
> > >
> > > As noted by our previous response (point 3), we have been working on applying our method on the PSPNet-50 architecture, and now we post our preliminary results. We adopt a PSPNet from https://github.com/hszhao/semseg to our current codebase. The results are shown in the table below. The result for the baseline PSPNet-50 is different from the original repo because we inherited HRNet's training recipe. The main difference seems to be in data augmentation. We will resolve the differences in our next experiments, and add the results to the paper's next version.
> > >
> > > We observe a similar trend on PSPNet, as our original HRNet experiments. Redesigned heads (RH) can improve early exit results by a large margin while confidence adaptivity (CA) helps reduce the FLOPs while maintaining decent accuracy. Our full method (RH + CA) saves 44.8% FLOPs compared with the original PSPNet (197.0 vs. 356.6 GFLOPs) with only 0.17% mIoU loss.
> > >
> > >
> > > |   &nbsp;&nbsp;&nbsp;&nbsp;&nbsp;&nbsp;&nbsp;&nbsp;&nbsp;Setting (PSPNet-50)   |               &nbsp;&nbsp;&nbsp;&nbsp;&nbsp;&nbsp;&nbsp;&nbsp;&nbsp;&nbsp;&nbsp;&nbsp;&nbsp;&nbsp;&nbsp;mIoU               | mean mIoU |           &nbsp;&nbsp;&nbsp;&nbsp;&nbsp;&nbsp;&nbsp;&nbsp;&nbsp;&nbsp;GFLOPs            | mean GFLOPs |
> > > | :---------------------: | :------------------------------: | :-------: | :-------------------------: | :---------: |
> > > |        Original (Baseline)        |      - &nbsp;&nbsp;&nbsp;&nbsp;&nbsp;&nbsp;&nbsp; -  &nbsp;&nbsp;&nbsp;&nbsp;&nbsp;&nbsp;&nbsp;&nbsp; -&nbsp;&nbsp;&nbsp;&nbsp;&nbsp;&nbsp;&nbsp;	73.07       |     -     |   &nbsp;-&nbsp;&nbsp;&nbsp;&nbsp;&nbsp;&nbsp; - &nbsp;&nbsp;&nbsp;&nbsp;&nbsp;&nbsp;-  &nbsp;&nbsp;&nbsp;&nbsp;&nbsp;&nbsp;356.6   |    122.7    |
> > > |      Early Exiting (Baseline)      | 29.91	44.40	63.84	72.90 |   52.76   |   22.4  34.3  95.8  351.0   |    125.9    |
> > > |   Early Exiting + RH (Ours)    | 38.32	46.16	64.93	73.11 |   55.63   |   20.5  30.5  84.0  339.2   |    118.5    |
> > > | Early Exiting + RH + CA (Ours) | 38.42	46.01	64.56	72.90 |   55.47   | 20.5  29.1  68.5  197.0 |  78.8  |

---

### Official Review · Reviewer_ohPA · 2021-11-02

**Correctness:** 3
**Technical Novelty And Significance:** 3
**Empirical Novelty And Significance:** 2
**Recommendation:** 6
**Confidence:** 4

**Details Of Ethics Concerns:**

No ethical concern spotted.


**Main Review:**

Strengths:
+ The task setup is interesting and of great practical value, as well-motivated in the paper.
+ The proposed method is simple, and makes intuitive sense, also the results have shown that it maintains a better accuracy-computation tradeoff, at least with the theoretic FLOPs measure.
+ The authors have open-sourced their implementation.
+ The writing is very clear and easy to follow.

Weaknesses:
- First, I appreciate that the authors discussed the matter of FLOPs vs runtime on page 7. Though, I'm a bit disappointmented about the fact that the proposed method does not achieve wallclock speedup with GPUs. As for author's argument about the potential of the proposed method with more development on sparse convolution, it holds with the assumption that it would be a cost-effective route to invest more into software/hardware support for sparse convolution, compared to investing more on developing models that are more friendly to batch computation and parallel hardware. In short, I'd be more convinced if I can see some wallclock acceleration from the proposed approach, even if it's far lower than its theoretical FLOPs speedup due to hardware constraints.
- An important ablation that's missing is for the interpolating neighborhood size gamma, as I'd imagine it has a large effect on interpolating quality vs computation trade-off. In fact, I cannot find in the paper what's the default value used in the experiments.
- It's not clear from the texts whether the confidence adaptivity mechanism is used in both training and testing? Or only during testing? It would be interesting to show results for both settings.
- Though it's shown as a visualization on specific samples (Figure 4), it would be helpful to also show quantitatively the portion of pixels that exceed thresholds at each stage.
- Cityscapes is relatively small and only with limited number of classes, it would make the results more convincing if the authors could repeat their experiments also on large datasets like ADE.
- Table 1, when comparing "Early Exiting + RH (HRNet)" and "Early Exiting + RH + CA (HRNet)", the gain mainly comes from exit 1, in fact if you average exit 2, 3, 4, you will get almost the same accuracy (72.1 vs 72.2), does this mean that the utility of RH is mainly to provide more layers for exit 1 so that it can produce a high enough modeling capacity?
- Page 7, "our model with confidence values as mask indicators also outperforms the feature-based mask sampling method in accuracy-computation tradeoff", when considering comparing to feature-based mask sampling method, shouldn't the correct comparison be "feature-based mask sampling" vs. "Early Exiting (HRNet)"? As my understanding is that "feature-based mask sampling" is not using an adaptive head for each exit and thus not a fair comparison to "Early Exiting + RH (HRNet)".

Questions/concerns:
- For the re-designed head, the authors mention that only 1x1 conv is used for both the encoder and decoder, would this be too limiting? Especially for earlier exits where the receptive fields of features are not large enough.
- Table 1, for the entry "Early Exiting (HRNet)", it's not clear what does the head design look like in this case? Are they with similar parameter counts?
- Table 1, as I mentioned before, it's not clear whether the entry "Early Exiting + RH + CA (HRNet)" use CA both in training and testing?

Typos:
- Page 2, "multiple predictors branch of from" --> "branch off"



**Summary Of The Paper:**

This paper proposes an anytime method for pixel recognition tasks like semantic segmentation and human pose estimation. The key idea is to design a network with multiple "early exits", from which the network could make predictions using a corresponding prediction head. Thus, depending on criteria like budget or confidence, the network could decide where to exit and therefore leads to different accuracy-computation operating points (i.e. its anytime property).

The technical contributions are mainly on two parts. The first contribution is a re-design of prediction heads (denoted as RH in the paper) at each early exits, mainly to tackle the difference of granularity for different intermediate feature maps. Another contribution is a confidence-based adaptive filtering mechanism that decides how to allocate computation budgets across spatial regions. Specifically, at each exit, the max prediction scores are used as a measure such that all spatial pixels that exceeds certain score threshold will not be processed in later stages.

The authors evaluated their methods on two tasks (Cityscapes for semantics segmentation and MPII for human pose estimation) and demonstrated that their full-blown method yields a better accuracy-computation tradeoff, compared to variants of the proposed approach and several previous methods.

------------------------- POST REBUTTAL -------------------------

The authors have addressed most of my concerns in their response. Also, after some discussions on the issue of lacking positive signals from wallclock time metric, I buy the arguments to position this work as a forward looking exploration in the alternative space to methods that are chosen in current hardware lottery. With this, I will raise my rating to positive inclined.





**Summary Of The Review:**

The proposed approach is simple and has nice accuracy-FLOPs tradeoff. However, the proposed approach does not seem to yield actual wallclock speedup due to its incompatibility with current parallel chips (GPUs). I'm not an expert in HPC, but I'm not sure about the assumption that this will change in near future. Without much positive signal on that, I'm doubtful about this general direction.

---

> ### Author Response · Authors · 2021-11-23
> **Response to Reviewer ohPA [1/3]**
>
> We sincerely thank you for your constructive comments. We are encouraged that you appreciate the practical value of the task, the simplicity and intuition of our method, and find the paper well-written. We would like to address the comments and questions below.
>
> 1. > ...method does not achieve wallclock speedup with GPUs... it holds with the assumption that it would be a cost-effective route to invest more into software/hardware support for sparse convolution, compared to investing more on developing models that are more friendly to batch computation and parallel hardware... In short, I'd be more convinced if I can see some wallclock acceleration from the proposed approach...
>
> **We believe research and engineering effort in sparse operations and developing models that are more friendly to batch/parallel computation are not contradictory, but complementary.** For example, a model that can be run with massively parallel computation can also possibly benefit from accelerated sparse operations.
>
> **We have surveyed a few recent works on sparse operations in neural networks, looking to apply them to our method.** SparseRT [1] accelerates sparse linear algebra operations utilizing unstructured (weight-level) sparsity. NVIDIA's TensorRT [2] achieves speedup on convolutions with sparse weights. Gale et al. [3] implement kernels for sparse matrix-matrix multiplication (SpMM) and sampled dense–dense matrix multiplication (SDDMM), achieving speedup on Transformers and MobileNets. Efficient modules with sparse input feature maps and/or weights are supported and tested on FPGA in [4]. Minkowski Engine [5] is also a popular library for ConvNets with sparse input and/or weights. These approaches [1-5] accelerate convolutions with sparse kernel weights and/or input feature maps, while our approach's sparsity is in a convolution's output feature maps. Elsen et al. [6] develop an efficient module for channel-wise sparse weights, while our sparsity is spatial. Therefore, unfortunately, **these implementations [1-6] do not directly fit in our method and cannot be directly used.**
>
> **Similar to our method, Figurnov et al. [7] and Xie et al. [8] also operate with sparse output feature maps without real wall-clock time savings yet. As discussed above, sparse computation in neural networks is still an ongoing and active research area, and we believe our method could benefit from future methods.**
>
> [1] SparseRT: Accelerating unstructured sparsity on GPUs for deep learning inference. Wang et al. 2020. International Conference on Parallel Architectures and Compilation Techniques \
> [2] Accelerating sparse deep neural networks. Mishra et al. 2021. arXiv:2104.08378. \
> [3] Sparse GPU kernels for deep learning. Gale et al. 2020. International Conference for High Performance Computing, Networking, Storage and Analysis. \
> [4] SparTen: A sparse tensor accelerator for convolutional neural networks. Gondimalla et al. 2019. International Symposium on Microarchitecture. \
> [5] Minkowski Engine. https://github.com/NVIDIA/MinkowskiEngine \
> [6] Fast sparse ConvNets. Elsen et al. CVPR 2020. \
> [7] Spatially Adaptive Computation Time for Residual Networks. Figurnov et al. CVPR 2017. \
> [8] Spatially Adaptive Inference with Stochastic Feature Sampling and Interpolation. Xie et al. ECCV 2020.
>
> 2. > An important ablation that's missing is for the interpolating neighborhood size gamma, as I'd imagine it has a large effect on interpolating quality vs computation trade-off. In fact, I cannot find in the paper what's the default value used in the experiments.
>
> Thank you for pointing to this important ablation. **The default value used is 7. We analyzed the sensitivity to the interpolation radius on HRNet-w18 below**:
>
> | &nbsp; Radius |  &nbsp; &nbsp;&nbsp; mIoU (four exits)     | Mean mIoU | GFLOPs (four exits)                     | Mean GFLOPs |
> | :-----------: | :--------------------------------: | :---------: | :--------------------------: | :-----------: |
> | 3           | 41.06	48.25	67.56	75.82 | 58.17     | 23.7 33.0  44.4  57.0 | 39.5       |
> | 5           | 40.86	48.01	67.64	76.21 | 58.18     | 23.7  33.1  44.6  57.4 | 39.7       |
> | 7 (default) | 41.05	48.35	67.73	76.10 | 58.31     | 23.7  33.1  44.9  58.1 | 40.0       |
> | 9           | 41.01	48.41	67.88	76.08 | 58.35     | 23.7  33.2  45.4  59.1 | 40.4       |
>
> It can be seen that mIoUs are not highly sensitive to the radius value, and 7 is a reasonable choice, as further increasing to 9 does not provide material gains. We have updated the paper, clearly indicating that radius is set to 7 in Section 3, and included these new results and discussion in **Appendix A**.

---

> > ### Author Response · Authors · 2021-11-23
> > **Response to Reviewer ohPA [2/3]**
> >
> > 3. > It's not clear from the texts whether the confidence adaptivity mechanism is used in both training and testing? Or only during testing? It would be interesting to show results for both settings.
> >
> > **It is used in both training and testing in experiments.** We have updated the paper indicating it is used in both at the end of Section 3. **In the table below, we compare confidence adaptivity at both training and inference, with "inference-only"**, taking an HRNet-W48 with redesigned heads (RH):
> >
> > | &nbsp; &nbsp; &nbsp;&nbsp;&nbsp; &nbsp;   Adaptivity | &nbsp; &nbsp;&nbsp; mIoU (four exits)                      | Mean mIoU | GFLOPs (four exits)                     | Mean GFLOPs |
> > | :-------------------: | :--------------------------------: | :---------: | :--------------------------: | :-----------: |
> > | No Adaptivity            | 44.61   60.19   76.64   79.89 | 65.33     | 41.9  105.6 368.0 701.4 | 304.2     |
> > | Training and Inference          |  44.34	60.13	76.82	81.31  | 65.7 | 41.9 93.9 259.4 387.1 | 195.6 |
> > | Inference-only | 44.61   59.97   76.37   79.69 | 65.16     | 41.9  94.1  291.8 484.8 | 228.1      |
> >
> > We can see that "inference-only" is worse than adaptivity at both training and testing. We have added this experiment and discussion to **Appendix B**.
> >
> > 4. > Though it's shown as a visualization on specific samples (Figure 4), it would be helpful to also show quantitatively the portion of pixels that exceed thresholds at each stage.
> >
> > Since the portion of pixels varies for different images, **we list the average percentage of pixels exceeding the threshold at each exit below** (HRNet-W48, Cityscapes).
> >
> > | exit 1 | exit 2 | exit 3 |
> > | ------ | ------ | ------ |
> > | 19.3% | 38.4% | 63.0% |
> >
> > The ratio is increasing as we go deeper, and a majority of pixels are masked after exit 3. We added this data in the revision.
> >
> > 5. > Cityscapes is relatively small and only with limited number of classes, it would make the results more convincing if the authors could repeat their experiments also on large datasets like ADE.
> >
> > **We added results on the PASCAL Context dataset [1], a dataset of 59 classes (much larger than Cityscapes' 19 classes) in Appendix C.** Please see results and discussions in the revision.  We will include results on ADE20K in our next version.
> >
> > [1] The Role of Context for Object Detection and Semantic Segmentation in the Wild. Mottaghi et al. CVPR 2014.
> >
> > 6. > Table 1, when comparing "Early Exiting + RH (HRNet)" and "Early Exiting + RH + CA (HRNet)", the gain mainly comes from exit 1, in fact if you average exit 2, 3, 4, you will get almost the same accuracy (72.1 vs 72.2), does this mean that the utility of RH is mainly to provide more layers for exit 1 so that it can produce a high enough modeling capacity?
> >
> > **We would like to note that the first early exit does not cost a large amount of computation.** For example, in HRNet-W48, our first early exit only costs 2.08 GFlops, while the previous calculations (network before the first exit) cost 41.92 GFlops. The low GFlops is partly due to aggressive downsampling operations and the use of 1x1 convolutions. We believe the increase of mIoU comes mainly from the increased receptive fields rather than added computation. In fact, **the redesigned heads have fewer parameters/FLOPs than the original heads** (see our response to point 9 for details).
> >
> > 7. > Page 7, "our model with confidence values as mask indicators also outperforms the feature-based mask sampling method in accuracy-computation tradeoff", when considering comparing to feature-based mask sampling method, shouldn't the correct comparison be "feature-based mask sampling" vs. "Early Exiting (HRNet)"? As my understanding is that "feature-based mask sampling" is not using an adaptive head for each exit and thus not a fair comparison to "Early Exiting + RH (HRNet)".
> >
> > In this work, we adapt the feature-based mask sampling method to work with multiple early exits for the anytime prediction task (details in "Baselines" paragraph of Section 4). **This is "adaptive" in the sense that the masks are generated from features, which are different for different inputs. We aim to compare this technique from prior work (adapted to the Early Exiting framework for anytime prediction) to our new contributions, namely redesigned heads and confidence adaptivity, hence the comparison.**
> >
> > In addition, we would like to also clarify that we do not view "Early Exiting" our new contribution, as it is commonly used by prior works in image classification as a framework. Therefore, in the revision, we moved "Early Exiting (HRNet)" from "ours" to "baselines" in Table 1. We hope this can help clarify our message.

---

> > > ### Author Response · Authors · 2021-11-23
> > > **Response to Reviewer ohPA [3/3]**
> > >
> > >
> > > 8. > For the re-designed head, the authors mention that only 1x1 conv is used for both the encoder and decoder, would this be too limiting? Especially for earlier exits where the receptive fields of features are not large enough.
> > >
> > > In encoders, **1x1 convs are used with 2x2 poolings alternately to enlarge the receptive fields** (as depicted in Fig. 2). In our early exploration, we found this to be on par with consecutive convs with kernel size 3 and stride 2. The choice of using 1x1 convs is also motivated by designing lightweight exit heads, in terms of added parameters and FLOPs.
> > >
> > > 9. > Table 1, for the entry "Early Exiting (HRNet)", it's not clear what does the head design look like in this case? Are they with similar parameter counts?
> > >
> > > **The head for "Early Exiting (HRNet)" is two consecutive 1x1 convolutions (with activation in between), following HRNet's original design.** We added this information in the experiment section. **Our redesigned heads actually have fewer parameters/FLOPs than the original heads, due to the operation (fixing channel width) described at the end of the "Head Redesign" part in Section 3.** For parameters, original heads in total take 0.22M parameters, our redesigned heads take 0.20M parameters. For FLOPs, please refer to Table 1 "Early Exiting" and "Early Exiting + RH" -- FLOPs decreased slightly with redesigned heads. Compared to the whole network, both head designs only contribute less than 0.3% parameters.
> > >
> > > 10. > Page 2, "multiple predictors branch of from" --> "branch off"
> > >
> > > We corrected the typo in the revision. Thanks for catching it.
> > >
> > > We would like to thank the reviewer again and we hope our response and revised submission could address your concerns.

---

### Official Review · Reviewer_GYrC · 2021-11-02

**Correctness:** 3
**Technical Novelty And Significance:** 3
**Empirical Novelty And Significance:** 3
**Recommendation:** 6
**Confidence:** 4

**Main Review:**

The proposed method is thoughtful and demonstrates good empirical performance for anytime pixel-level recognition. This paper is also very well-written.

The proposed method is not friendly for hardwares like GPU. But I like the part where the authors implement their method and demonstrate speedup on the CPU. One small clarification: what’s the metric used here to evaluate the computation on CPU? Is that the end-to-end latency?

My main concern is its technical novelty. Early exits have been explored in many previous work, e.g. Multi-scale DenseNet for anytime image classification (Huang et al., 2017). While pixel-level recognition is definitely more challenging than image classification, the solution proposed in this work is still similar to the spirit of previous work.

Also, it seems that most gain brought by the proposed method comes from the “confidence adaptivity” strategy. I find the interpolation idea really interesting and promising. But as mentioned by the authors, this strategy was originally proposed by Xie et al., 2020 (ECCV). So this downgrades the contribution of this work.

In Table 1, why do we need to show the average performance across exits? I feel the curve in Figure 3 would provide a more complete picture of the performance & efficiency. But the average performance is not really informative here.

Does the method generalize to other architectures other than HRNet? It would be helpful to see the proposed idea can work on more than one architecture family (even though HRNet is the current SOTA for segmentation).

**Summary Of The Paper:**

This work focuses on anytime pixel-level recognition (e.g., semantic segmentation). They propose to add intermediate exists in the architecture for anytime inference. They also consider spatial confidence adaptivity in their network, where they only execute subsequent layers on a small set of non-confidence pixels and obtain the features of other positions via interpolation. They apply the method to semantic segmentation and human pose estimation and demonstrate a reduction in FLOPs and good anytime performance for both tasks.


**Summary Of The Review:**

My main concern is that the technical novelty of this work is somewhat limited. Other than that, this work is solid and has comprehensive results and analysis. So, I feel this work is borderline and am slightly towards borderline accept.

---

> ### Author Response · Authors · 2021-11-23
> **Response to Reviewer GYrC**
>
>
> We sincerely thank you for your constructive comments. We are encouraged that you find the method thoughtful, the empirical performance good, and the paper well-written. We would like to address the comments and questions below
>
> 1. > The proposed method is not friendly for hardware like GPU. But I like the part where the authors implement their method and demonstrate speedup on the CPU.
>
> We are glad you find our CPU experiment valuable. We acknowledge that sparse operations are not GPU-friendly in general. **For a more detailed discussion on GPU acceleration, please refer to our response (point 3) to Reviewer hS3f.**
>
> 2. > what’s the metric used here to evaluate the computation on CPU? Is that the end-to-end latency?
>
> **Yes, it is the end-to-end latency**. We have updated our paper accordingly. Thank you for pointing this out.
>
> 3. > My main concern is its technical novelty. Early exits have been explored in many previous work ... this work is still similar to the spirit of previous work.
>
>
> As noted by the reviewer and our Section 3, Early Exiting (EE) has been used by prior works, mainly in classification tasks. We believe it is still important to see how such a framework performs at pixel-level tasks.
>
> In addition, **we would like to clarify that we do not view EE as our new contribution or technical novelty**. Our contributions include the **confidence adaptivity mechanism**, which outperformed the feature-based stochastic sampling method in Xie et al.; the **redesigned exits**, which improved early exit results significantly; and **experimental results on recent alternatives** such as deep equilibrium networks (MDEQ), and feature-based stochastic sampling. This was indeed not communicated clearly in our original submission. **We have updated our paper, moving the EE result row in Table 1 from "Ours" to "Baselines", merging "Anytime Setting" and "Early Exiting" paragraphs in Section 3, and mentioning again EE was used in prior works before we introduce it.** We hope this can help make our contributions more clear.
>
> 4. > (The interpolation) strategy was originally proposed by Xie et al., 2020 (ECCV)
>
> **Interpolation is indeed a necessary part of our method.** We apply the same procedure as in Xie et al. (as cited) and find it is already effective. But **the interpolation technique is not sufficient by itself to achieve our results**.
>
> 5. > In Table 1, why do we need to show the average performance across exits?
>
> Yes, we agree that the curves portray a more complete picture for the accuracy-computation tradeoff. **The average statistics serve as simple aggregation of metrics for multiple exits.** For example, Fig. 5 (right) uses the average FLOPs and accuracies for comparing different masking criteria. **We could remove the averages in Table 1. in the next version if needed.**
>
>
> 6. > Does the method generalize to other architectures other than HRNet?
>
> **We chose HRNet as our backbone because it is (relatively) simple, effective, and quite widely used. As noted by the reviewer, the current state-of-the-art methods on Cityscapes are still based on HRNet** (e.g., see https://paperswithcode.com/sota/semantic-segmentation-on-cityscapes). We are currently experimenting with PSPNet [1] to test our method on another architecture. We will add the result to our next version, and post an update comment if we manage to get the results before the discussion period ends (Nov 29). PSPNets are quite different from HRNets so it will take some time and effort. We also welcome suggestions on which ConvNet-based architecture(s) to further test our method on.
>
> [1] Pyramid scene parsing network. Zhao et al. CVPR 2017.
>
> We thank you again for your valuable feedback and we hope our response and updated submission can address your concerns.

---

> > ### Author Response · Authors · 2021-11-29
> > **More architecture result update**
> >
> > As noted by our previous response (point 6), we have been working on applying our method on the PSPNet-50 architecture, and now we post our preliminary results. We adopt a PSPNet from https://github.com/hszhao/semseg to our current codebase. The results are shown in the table below. The result for the baseline PSPNet-50 is different from the original repo because we inherited HRNet's training recipe. The main difference seems to be in data augmentation. We will resolve the differences in our next experiments, and add the results to the paper's next version.
> >
> > We observe a similar trend on PSPNet, as our original HRNet experiments. Redesigned heads (RH) can save improve early exit results by a large margin while confidence adaptivity (CA) helps reduce the FLOPs while maintaining decent accuracy. Our full method (RH + CA) saves 44.8% FLOPs compared with the original PSPNet (197.0 vs. 356.6 GFLOPs) with only 0.17% mIoU loss.
> >
> >
> > |   &nbsp;&nbsp;&nbsp;&nbsp;&nbsp;&nbsp;&nbsp;&nbsp;&nbsp;Setting (PSPNet-50)   |               &nbsp;&nbsp;&nbsp;&nbsp;&nbsp;&nbsp;&nbsp;&nbsp;&nbsp;&nbsp;&nbsp;&nbsp;&nbsp;&nbsp;&nbsp;mIoU               | mean mIoU |           &nbsp;&nbsp;&nbsp;&nbsp;&nbsp;&nbsp;&nbsp;&nbsp;&nbsp;&nbsp;GFLOPs            | mean GFLOPs |
> > | :---------------------: | :------------------------------: | :-------: | :-------------------------: | :---------: |
> > |        Original (Baseline)        |      - &nbsp;&nbsp;&nbsp;&nbsp;&nbsp;&nbsp;&nbsp; -  &nbsp;&nbsp;&nbsp;&nbsp;&nbsp;&nbsp;&nbsp;&nbsp; -&nbsp;&nbsp;&nbsp;&nbsp;&nbsp;&nbsp;&nbsp;	73.07       |     -     |   &nbsp;-&nbsp;&nbsp;&nbsp;&nbsp;&nbsp;&nbsp; - &nbsp;&nbsp;&nbsp;&nbsp;&nbsp;&nbsp;-  &nbsp;&nbsp;&nbsp;&nbsp;&nbsp;&nbsp;356.6   |    122.7    |
> > |      Early Exiting (Baseline)      | 29.91	44.40	63.84	72.90 |   52.76   |   22.4  34.3  95.8  351.0   |    125.9    |
> > |   Early Exiting + RH (Ours)    | 38.32	46.16	64.93	73.11 |   55.63   |   20.5  30.5  84.0  339.2   |    118.5    |
> > | Early Exiting + RH + CA (Ours) | 38.42	46.01	64.56	72.90 |   55.47   | 20.5  29.1  68.5  197.0 |  78.8  |

---

> > > ### Comment · Reviewer_GYrC · 2021-11-29
> > > **Response to the rebuttal**
> > >
> > > Thanks for providing the rebuttal and extra results on PSPNet-50. The response has resolved my concerns. I will keep my previous positive rating.

---

> > > > ### Author Response · Authors · 2021-11-29
> > > > **Response to Reviewer GYrC**
> > > >
> > > > We are glad that we addressed your concerns in the rebuttal. Thank you again for your feedback and suggestions.

---

### Official Review · Reviewer_hS3f · 2021-11-02

**Correctness:** 3
**Technical Novelty And Significance:** 3
**Empirical Novelty And Significance:** 3
**Recommendation:** 8
**Confidence:** 3

**Main Review:**

**Strengths**

i) Submission includes well-documented code.

The code is included in submission and the paper has an anonymous github link. README.md describes setup steps.  The code itself is reasonably well-organized.  Though I did not see many comments or docstrings, the names are reasonable and there are at least some logging messages that help make it clear what's going on in parts. Provided sample commands seem to work (at least as far as training).

Minor comment: Sample commands appear to use the authors' specific GPU setup, this seems unnecessary. It's good to have simple README sample commands targeting the least common denominator to debug environment setup and dependency installation. Though the minimal command may not be that simple due to calls to torch.distributed in the source.

ii) Description of adaptive/early exit method is overall very clear.

Figure 2 makes the high-level idea very clear, showing where the early exits and masking is done. The novel modules are described precisely in mathematical notation in the "Confidence Adaptivity" subsection.

iii) Good FLOPs/accuracy tradeoff experiments.

Experiments look at the pareto frontier of computational cost (as FLOPs) vs accuracy (as mIOU or probability of correct keypoint). They include a reasonable baseline, MDEQ that also varies in the tradeoff. Results are good, showing that the authors' method dominates the baseline.

Minor Point: Varying the HRNet architecture for mulitple tradeoff points also seems like a good baseline to illustrate here, rather than plotting only the single backbone used.

**Weaknesses**

iv) Relatively expensive early exits.

The early exits must have multiple layers of downsampling & upsampling, as described in the "Head Redesign" subsection. It does seem necessary to do this apply this to segmentation, and the authors give a good description of the reasoning. This "redesign" seems like an important novelty of the submission. The heavyweight early exists maybe make this harder to apply than the very simple early exits in classification work such as Veit & Belongie. The authors do still show some computational savings, though, so this extra cost is clearly not severe enough to invalidate the method.

v) Some unclear parts of interpolation.

The description of the interpolation, in the span between and including equations (2)-(3), does not make the handling of the mask that clear. Will zeroed-out features be included in the average for I(f_{out})? The weighting in (3) seems to only take distance into acount, not the value of the mask.

vi) Limited empirical handling of wall-clock time.

The submission describes the considerations behind the decision to report FLOPS at the bottom of page 7. The authors correctly note that reporting FLOPs is common, and that this is due to the need for lots of work on the acceleration of sparse computations in underlying libraries and hardware (see also "The Hardware Lottery" by Hooker). But the most recent work on this that they cite, such as Elsen et. al. 2020, may be worth considering in experiments: it is fairly well-optimized and has a mature implementations. So we'd expect that the best possible likely speedups in the near future are at least not that far from what would be achieved with the implementations of the past 1-2 years. In some experiments, such as the HRNet-W48 network on Cityscapes, the advantage of early exiting looks slim enough that it seems doubtful one would see wall-clock speedups even with fairly good acceleration of the sparse computations.

The video included in the submission also illustrates the real timings, and shows that the wall-clock time is probably at least reasonable: but this is less useful than quantitative experiments.

**Minor Comments**

  * "this strategy work well" -> "this strategy works well" at bottom of page 3.
  * It's usually better to number *all* equations, for instance on page 5. This makes it easier for readers (and reviewers!), or paper citing this one, to reference specific equations, even if that equation isn't cross-referenced in the paper itself.

**Summary Of The Paper:**

The paper presents adaptive and anytime methods for semantic segmentation and pose recognition. Both of these are "pixel-level" tasks where the model is expected to output a prediction for each pixel in the input image. Their adaptive method, that performs variable amounts of computation depending on the input and the budget, with the amount of computation devoted to a given pixel varying from one part of the image to another. This is done by adding early exits to the base model architecture, and modifying the convolution layers to perform sparse computation on a subset of locations followed by interpolation. Experiments are done on Cityscapes semantic segmentation and MPII pose estimation benchmarks.

**Summary Of The Review:**

The method is interesting and presented well. Weaknesses in presentation and experiments are mostly lower-level and affect only parts of the method and its description or validation, and are not central to the overall paper.

---

> ### Author Response · Authors · 2021-11-23
> **Response to Reviewer hS3f**
>
> We sincerely thank you for your constructive comments. We are encouraged that you find our method description clear, our results good, and our code well-documented. We would like to address the comments and questions below.
>
> 1. > iv) Relatively expensive early exits
>
> As shown in Table 1, our "RH" model takes 701.3 GFLOPs, only a very small percentage (<1%) addition compared to vanilla HRNet's 696.2 GFLOPs. Therefore **these early exits are not that expensive in terms of computation**. This is partly because we use 1x1 convolutions and 2x2 pooling in the exits, which are lightweight in parameters and FLOPs, and also the operation (fixing channel width) described at the end of the "Head Redesign" part in Section 3.
>
>
> 2. >  v) Some unclear parts of interpolation. Will zeroed-out features be included in the average for I(f_{out})?
>
> **Yes, zeroed-out features will still participate as the input of the interpolation process, where they take values of zero**. Empirically we found this works reasonably well. We have added "Note that masked-out features ($M_i(p) = 0$) still participate in the interpolation process as inputs with values of 0" in the revision to make this more clear.
>
> 3. > vi) Limited empirical handling of wall-clock time... the most recent work on this that they cite, such as Elsen et. al. 2020, may be worth considering in experiments: it is fairly well-optimized and has a mature implementations ...
>
>
> **We have surveyed a few recent works on sparse operations in neural networks, looking to apply them to our method.** SparseRT [1] accelerates sparse linear algebra operations utilizing unstructured (weight-level) sparsity. NVIDIA's TensorRT [2] achieves speedup on convolutions with sparse weights. Gale et al. [3] implement kernels for sparse matrix-matrix multiplication (SpMM) and sampled dense–dense matrix multiplication (SDDMM), achieving speedup on Transformers and MobileNets. Efficient modules with sparse input feature maps and/or weights are supported and tested on FPGA in [4]. Minkowski Engine [5] is also a popular library for ConvNets with sparse input and/or weights. These approaches [1-5] accelerate convolutions with sparse kernel weights and/or input feature maps, while our approach's sparsity is in a convolution's output feature maps. Elsen et al. [6] develop an efficient module for channel-wise sparse weights, while our sparsity is spatial. Therefore, unfortunately, **these implementations [1-6] do not directly fit in our method and cannot be directly used.**
>
> **Similar to our method, Figurnov et al. [7] and Xie et al. [8] also operate with sparse output feature maps without real wall-clock time savings yet. As demonstrated above, sparse computation in neural networks is still an ongoing and active research area, and we believe our method could benefit from future methods.**
>
> [1] SparseRT: Accelerating unstructured sparsity on GPUs for deep learning inference. Wang et al. 2020. International Conference on Parallel Architectures and Compilation Techniques \
> [2] Accelerating sparse deep neural networks. Mishra et al. 2021. arXiv:2104.08378. \
> [3] Sparse GPU kernels for deep learning. Gale et al. 2020. International Conference for High Performance Computing, Networking, Storage and Analysis. \
> [4] SparTen: A sparse tensor accelerator for convolutional neural networks. Gondimalla et al. 2019. International Symposium on Microarchitecture. \
> [5] Minkowski Engine. https://github.com/NVIDIA/MinkowskiEngine \
> [6] Fast sparse ConvNets. Elsen et al. CVPR 2020. \
> [7] Spatially Adaptive Computation Time for Residual Networks. Figurnov et al. CVPR 2017. \
> [8] Spatially Adaptive Inference with Stochastic Feature Sampling and Interpolation. Xie et al. ECCV 2020.
>
> 4. > Minor comment: Sample commands appear to use the authors' specific GPU setup, this seems unnecessary.
>
> We have updated the repo README page and simplified the commands.
>
> 5. > Minor Point: Varying the HRNet architecture for multiple tradeoff points also seems like a good baseline to illustrate here, rather than plotting only the single backbone used.
>
> **We have experimented with two HRNets of different sizes**, HRNet-W48 and HRNet-W18 for the semantic segmentation task (Fig. 3 left and middle). The model we used for pose estimation is an HRNet-W32. These follow the original HRNet paper.
>
> 6. > "this strategy work well" -> "this strategy works well" at bottom of page 3.
>
> We have corrected this. Thank you for catching it.
>
> 7. > It's usually better to number all equations.
>
> The suggestion makes sense to us and we have numbered all equations in the revision.
>
> We thank you for the valuable feedback again and hope our response and updated submission can address your concerns.

---

> > ### Comment · Reviewer_hS3f · 2021-11-29
> > **Response to clarifications**
> >
> > 1. I had previously missed that the submission has exact FLOPS counts for the early exits. The early exits are reasonable, based on these numbers.
> > 2. Including the zeros is counter-intuitive, but there's no reason that it seems incorrect. The proposed clarification is good.
> >
> > 4-7. Minor points that seem to have been corrected.
> >
> > 3. Including wall-clock time results should not be a hard requirement for publication in this space. Doubly-so for wall-clock time on GPUs, since NN inference need not target that hardware, especially when considering efficient NN methods.
> >
> > But when work is ongoing in other research fields to implement the necessary hardware & software to support methods similar to the submission, at some point these should converge. So a paper with wall-clock times is stronger than one without. I'd argue this is the case even if it reports negative results on wall-clock time, though I recognize the reality of how this'd be received by many reviewers.
> >
> > Still, this shouldn't be a hard requirement for ICLR, and the paper has novel contributions that have been validated to the extent possible based on "idealized" (if unrealistic) measures. The quality and future potential usefulness of the work should be the metric, not whether it's a present-day winner of the "hardware lottery." (Coincidentally, a version of this essay has been published in CACM during the discussion period: https://cacm.acm.org/magazines/2021/12/256929-the-hardware-lottery/fulltext, I think this is an important read for background to the review discussion for this paper.)
> >
> > So if the primary weakness is the lack of wall-clock time results in the paper, I think this should lead to a decision to accept.

---

> > > ### Author Response · Authors · 2021-11-29
> > > **Response to Reviewer hS3f**
> > >
> > > Thank you for your response to our clarifications, and we are glad you find most points clarified. The pointed Communications of ACM article ("The Hardware Lottery") is indeed insightful and interesting to us. It is also highly related to this work and other related works' situation on hardware compatibility. While we acknowledge the importance of wall-clock acceleration on present-day, mainstream hardwares for a method to be popularly used, we also fully agree with your and the article's view that it is important to choose research directions based on its potential future usefulness. We will add a brief discussion on the "Hardware Lottery" article in our next version.

---

### Author Response · Authors · 2021-11-28
**Invitation for discussions**

Dear reviewers,

We thank you for your valuable feedback and suggestions, which we used to update and improve our submission in the rebuttal process. We would like to kindly invite you to respond to our author rebuttal, so that we could answer your further questions or clarify unclear points, if there is any. In our responses, we added some new results as requested by the reviewers, clarified on the method’s technical novelty, and discussed about method’s FLOPs vs. wall-clock time. We hope our responses and paper revision could address your concerns. Thanks!

Best, \
Authors

---

### Comment · Reviewer_2wqr · 2021-11-29
**Post-Rebuttal Comments**

I have read the rebuttal, and I think the authors did a good job addressing most of my concerns. I still think that the paper is a bit technically incremental. However, given the authors' rebuttal response, and the positive comments of other reviewers, I'm willing to improve my initial rating to "accept" (as a poster).

---

> ### Author Response · Authors · 2021-11-29
> **Response to Reviewer 2wqr**
>
> We feel encouraged that most concerns have been addressed. Thanks again for for reading the author rebuttal and other comments.

---

### Comment · Reviewer_ohPA · 2021-11-29
**Post-rebuttal review**

The authors have addressed most of my concerns in their response. Also, after some discussions on the issue of lacking positive signals from wallclock time metric, I buy the arguments to position this work as a forward looking exploration in the alternative space to methods that are chosen in current hardware lottery. With this, I will raise my rating to positive inclined.

---

> ### Author Response · Authors · 2021-11-29
> **Response to Reviewer ohPA**
>
> We are glad to hear most of your concerns have been addressed. Thanks again for for reading the author rebuttal and other discussions.

---

### Decision · Program_Chairs · 2022-01-20

**Decision:**

Accept (Poster)

**Comment:**

This submission received 4 final ratings above the acceptance threshold: 6, 6, 6, 8. The reviewers mentioned limited novelty, but acknowledged practical importance of this work, and particularly appreciated thorough analysis provided by the authors. After a strong rebuttal, most of remaining concerns have been addressed.
The final recommendation is therefore to accept this submission as a poster.